



# 1   Oxycline oscillations induced by internal waves in deep Lake Iseo

Giulia Valerio[1], Marco Pilotti[1,4], Maximilian Peter Lau[2,3], Michael Hupfer[2]
[1.] DICATAM, Università degli Studi di Brescia, via Branze 43, 25123 Brescia, Italy
[2.] Leibniz-Institute of Freshwater Ecology and Inland Fisheries, Müggelseedamm 310, 12587
Berlin, Germany
[3.] Université du Quebec à Montréal (UQAM), Department of Biological Sciences, , Montréal, QC
H2X 3Y7, Canada
[4.] Civil & Environmental Engineering Department, Tufts University, Medford, MA 02155, USA
*Correspondence to*: Giulia Valerio (giulia.valerio@unibs.it)
**Abstract**
Lake Iseo is undergoing a dramatic de-oxygenation of the hypolimnion, representing an emblematic
example among the deep lakes of the prealpine area that are, to a different extent, suffering from
reduced deep water mixing. In the anoxic deep waters, the release and accumulation of reduced
substances and phosphorus from the sediments is a major concern. Since the hydrodynamics of this
lake was shown to be dominated by internal waves, in this study we investigate for the first time the
role of these oscillatory motions on the vertical fluctuations of the oxycline, currently situated at a
depth of around 95 m, where a permanent chemocline inhibits deep mixing by convection.
Temperature and dissolved oxygen data measured at moored stations show large and periodic
oscillations of the oxycline, with amplitude up to 20 m and periods ranging from 1 to 4 days. A deep
dynamics characterized by larger amplitudes at lower frequencies is shown to be favoured by the
excitation of second vertical modes in strongly thermally stratified periods and of first vertical modes
in weakly thermally stratified periods, when the deep chemical gradient can support baroclinicity
anyhow. These basin-scale internal waves cause in the water layer between 85 and 105 m depth a
fluctuation of the oxygen concentration between 0 and 3 mg L$^{-1}$ that, due to the bathymetry of the
lake, changes the redox condition at the sediment surface. This forcing, involving about 3% of the
lake's sediment area, can have major implications for the biogeochemical processes at the sediment
water interface and for the internal matter cycle.



## 1. Introduction

The physical processes occurring at the sediment-water interface of lakes are crucially controlling the fluxes of chemical compounds across this boundary (Imboden and Wuest, 1995), implying severe implications for water quality. In stratified lakes, the boundary-layer turbulence is primarily caused by wind-driven internal wave motions (Imberger, 1998). Consequentially, the periodicity of these large-scale oscillations is a likely cause of the unsteadiness of the sediment–water flux (Lorke et al., 2003).

A first reason of non-stationarity is the action of alternating velocity currents at the top of the benthic boundary layer (BBL), as theoretically explained by Jørgensen and Marais (1990) and Lorke and Peeters (2006). In the immediate vicinity of the water-sediments interface, the vertical transport of solutes occurs via molecular diffusion in diffusive sublayer. The thickness of this layer, which is solute-specific and in the order of few millimetres, depends strongly on the flow regime in the turbulent BBL above. Increasing levels of turbulence result in a compression of the diffusive sublayer and, according to Fick's first law, an increase of the solutes fluxes. The alternating currents are the main reason for transient variations in the sediment oxygen uptake rate and penetration depth as experimentally observed by Lorke et al. (2003), Brand et al. (2008) and Bryant et al. (2010) in Lake Alpnach, a 34 m deep lake known to feature pronounced seiching.

In thermally stratified lakes, a further driver for flux unsteadiness is the periodic occurrence of cyclic convective turbulence in the sediment area located in the water layer characterized by thermocline fluctuations. Here the sediments are exposed to pronounced temperature oscillations during internal seiches. In particular, during the upslope current, cold deep water flows over the warmer sediments. The resulting intermittent instability drives free convection, accelerating the fluxes at the sediment-water interface by more than one order of magnitude, as experimentally observed by Kirillin et al. (2009), Lorke et al. (2005) and Chowdhury et al. (2016). In lakes with anoxic water layers, the seiches-induced thermocline oscillations can be accompanied by periodical changes of oxygen in a large internal shoreline area during periods with complete hypolimnetic oxygen depletion as described by Deemer et al. (2015) for Lacamas Lake. Bernhardt et al. (2014) have observed similar seiches-induced oxygen fluctuations in the sediment-water interface in the shallower area of the eutrophic Lake Arendsee due to the formation of distinct metalimnetic oxygen minimum during summer.

These findings motivated us to explore if similar unsteady fluxes also occur in the much deeper waters of lakes where incomplete seasonal mixing creates a deep oxycline between the mixolimnion and a perennially stagnant and denser monimolimnion. The density gradient that is typically present across





these layers has been shown to support higher vertical baroclinicity (Salvadé et al., 1988; Roget et al.
2017), whose amplitude is typically larger than that of the thermocline. Accordingly, we hypothesize
that under the internal wave motions of the deep oxycline, the contiguous sediments undego
alternating redox conditions, with entailing implications for biogeochemical controlling the P fluxes
at the sediment-water interface. Although the oxygen gradient across such deep oxyclines (e.g. in
meromictic lakes) can be pronounced and typically persists beyond the seasonal stratification, field
investigation of the oxycline seiching remains, to our knowledge, missing. Using a three-layer model
of the Northern Lake Lugano, a 288 m deep meromictic lake characterized by a permanent
chemocline at 100 m, Salvadé et al. (1988) estimated oscillations of the chemocline, and hence of the
oxycline, up to 10 times larger than the thermocline. Hutter (2011) later invoked a field verification
of these computational results.
A suitable field site for this type of investigation is Lake Iseo, a deep lake where a chemocline at
around 95 m separates 4.7 km$^3$ of oxygenated waters (mixolimnion) and 3.2 km$^3$ of anoxic waters
(monimolimion). During the thermally stratified period, high-resolution temperature data (Pilotti et
al., 2013) highlighted a strong internal wave activity in the first 50 m, where the main ~25 h period
(V1H1 mode) is excited by the ordinary wind and is occasionally superimposed on a ~60 h period
(V2H1 mode), the latter being excited by long-lasting winds. The occurrence of these motions was
interpreted as the outcome of wind forcing with similar horizontal structures and with energies at
frequencies near the natural oscillations of the excited modes (Valerio et al., 2012). In this study, we
extended this analysis to the wind-induced movements of the deeper waters between 85 and 105 m,
where the oxycline is located, in order to provide an estimation of the spatial and temporal extent of
oxygen fluctuations at the surface of deep sediments. As deep sediments are generally known to be
potentially redox-sensitive phosphorus (P) sinks, we discuss our results in light of the expected P
fluxes from the contiguous sediments. The importance of this research is motivated by the observation
that Lake Iseo is currently undergoing a change in mixing pattern and P recycling, so that a deeper
understanding of the sediment P release dynamics is crucial to forecast the possible future trajectories
of this ecosystem.

**2. Methods**
**2.1 Field site**
Lake Iseo (see Fig. 1) is a 61 km$^2$ large and 256 m deep lake located in the pre-alpine area of Italy, at
the southern end of Valle Camonica, a wide and long glacial valley. In the first limnological study of



Lake Iseo, made in 1967, the lake was described as a monomictic and oligotrophic lake, featuring a
fully oxygenated water column and P concentrations of a few µg $L^{-1}$. Starting in the 1980s, the
accumulation of solutes from biomass processing, in combination with climatic factors, has gradually
inhibited deep water renewal. With decreased mixing depth, deep water quality deteriorated,
including increases in P concentrations and persistence of anoxic conditions (Garibaldi et al., 1999).
According to the profiles measured after the last winter (April 2018), the mixolimnion has raised to
a depth of 95 m, where the oxygen and conductivity profiles (see Fig. 2a) clearly highlight a sharp
chemocline at the same depth of the oxycline. The density gradient across this chemocline, calculated
at about 25 mg $L^{-1}$ (Pilotti et al., in preparation) seems to be sufficient, under current climatic
conditions, to prevent a deeper convective mixing. In the anoxic monimolimnion below, the amount
of P does not show any signs of decrease (currently with a space averaged concentration of ~111 µg
TP $L^{-1}$), and a recent field campaign has shown that the P stock is supplied to a comparable extent (~
15 tons of P $year^{-1}$) by both sedimentation from the layers above and by the fluxes from the sediment.

## 2.2 Field data

A wide set of experimental data were measured to describe the wind-induced movements of the
waters layers between the mixolimnion and monimolimnion of Lake Iseo at the lake stations shown
in Figure 1. We followed the meteorological and mechanical forcing at the lake surface in high
temporal resolution (60 s) by means of two on-shore stations measuring wind speed and direction, air
temperature, air humidity and short wave radiation (SS-1 and SS-2). Furthermore, a floating station
(LS-N) measured wind speed and direction and net long-wave radiation. LS-N is further equipped
with eleven submerged loggers that measure the temperature (± 0.01°C accuracy, 60 s interval), well
describing the vertical movements of the thermal gradient (see Fig. 2). In October 2017, we added
three additional temperature loggers at 55, 75 and 113 m depth, to better describe the temperature
fluctuations below the thermocline thanks to their higher accuracy (± 0.002°C).
To capture the vertical fluctuations of the oxycline, we installed four submersible instruments
(miniDOT, Precision Measurement Engineering, Vista, Ca, USA) at LS-S between 85 and 105 m,
measuring dissolved oxygen (DO) for nine consecutive months at a 1 $min^{-1}$ sampling frequency (see
Table 1). These loggers rely on a fluorescence-based oxygen measurement with an accuracy of ± 5%
of the measured value (mg $L^{-1}$). The LS-S logger chain was installed at a 105 m deep location in the
southern basin. Two additional loggers were installed in the northern station (LS-N) at 85 and 95 m.
As shown in Table 1, NO85 and NO95 measured the oxygen content at the same depths as the
southern instruments, but operated for a shorter period of time. In the following sections, we will



focus on the data analysis from July 2017 to February 2018, which fully captures the evolution of the
oxygen content during the transition from a strongly to a weakly stratified period.
On 21-22 July 2017, we also conducted a field campaign aimed at investigating the oxygen profiles
in the whole water column at higher vertical resolution. By means of a CTD probe, (RINKO CTD
profiler with an optical fast DO sensor,JFL Advantech Co. Ltd., Tokyo, Japan) we measured the
temperature and DO profiles alternatively at the two lake extremities in the proximity of the LS-N
and LS-S stations several times throughout the days. Similarly, consecutive DO profiles were
measured in the proximity of the LS-N stations on 10th, 16th and 18th of April 2018.

### 2.3 Numerical models

In this study, we used two numerical models to better highlight different dynamic aspects of the
measured internal oscillations. This required to identify the temporal evolution of the periodicity and
the spatial structure of the natural modes in Lake Iseo.
At a first stage, following the approach pursued in Guyennon et al. (2014) for Lake Como, a modal
analysis was performed to quantify the temporal evolution of the periods of the natural modes. This
model schematizes the density structure of a lake in layers of constant density and provides the free
baroclinic oscillations of the layers interfaces by solving an eigenvalue problem (details in Guyennon
et al., 2014). In the case of Lake Iseo, the bathymetry was discretized with a 160 x 160-m horizontal
grid, while the averaged vertical density structure was schematized with a 4 to 2 layers structure on a
monthly basis, as detailed in Table 2. As typical for the sub-alpine lakes, a pronounced 3-layers
thermal stratification forms in April, characterized by a well-mixed and warm surface layer, separated
from the cold hypolimnion by an intermediate metalimnion. The stronger thermal stratification is
reached in August. After the thermocline's deepening during the cooling period, the thermal
stratification reduces to 2 layers in winter time, separated by one interface between 35 and 55 m and
characterized by a weak thermal gradient, which finally disappears in March. In the case of Lake Iseo,
the thermal stratification is also superimposed to a chemical stratification. Thus, we considered an
additional deep layer separated from the hypolimnion by the chemocline at 95 m and characterized
by 25 mg L$^{-1}$ of additional density due to the higher concentration of dissolved salts. This value was
quantified on the basis of the chemical analysis for 2 water samples collected at a depth of 40 m and
200 m, according to the procedure proposed by Boehrer et al. (2010) (Pilotti et al., in preparation).
At a second stage, we determined in detail the vertical and horizontal structure of the natural modes
by means of  a 3D hydrodynamic model which accounts also for the non-linear terms of the



momentum equations. We made use of the hydrostatic version of the Hydrodynamic-Aquatic
Ecosystem Model (AEM3D, Hodges and Dallimore, 2016). This model was developed from the
ELCOM-CAEDYM model (Hodges et al. 2000) and was already successfully tested in simulating of
the internal wave activity in the upper 50 m of lake Iseo (Valerio et al., 2017). With respect to this
previous analysis, we also investigated the wind-induced oscillations around 100 m depth. For this
purpose, we thickened the vertical grid, imposing 150 vertical layers, 1 m thick, followed by layers
with gradually increased thickness up to 25 m for the deepest part of the lake. We also thickened the
horizontal grid, 80 x 80 m, to better describe the bathymetry in the southern and northern area. Finally,
we used a passive tracer to follow the vertical fluctuations of the oxycline forced by the wind. In order
to simulate the structure of each single mode of oscillation, we conducted numerical experiments in
which the lake was forced by a synthetic wind time series of sinusoidal form, with a maximum value
of 5 ms$^{-1}$, whose spatial and temporal structure fit the one the one predicted by the eigenmodel for
the natural modes of oscillation. This approach was already successfully applied by Vidal et al. (2007)
to the study of the higher vertical modes of Lake Beznar.

## 3. Results

### 3.1 Analysis of the measured data

The oscillatory motions measured around the thermocline and the oxycline show marked differences
in periodicities and amplitudes. This difference clearly stands out from the comparison of the 12°C
oscillations at LS-N (located between 13 and 30 m of depth in proximity to the thermocline depth
from July to November 2017, see Fig. 3a) and the 0.5 mg DO L$^{-1}$ oscillations at LS-S (located between
85 and 105 m of depth from July to February 2018, see Fig. 3c). To assess the frequency content of
these time series and their variability over time, we used the wavelet analysis. We applied the Morlet
transform to the two measured signals that, unlike the classical Fourier transform, allows a
localization of the signals in both frequency and time rather than in a simple frequency space
(Torrence and Compo, 1998).
Figure 3b (limited in time by the onset of the weakly stratified period, when the 12°C isotherms
becomes undetectable from our set of instruments) highlights a strong concentration of the energy of
the shallower oscillations around 1 day period during the whole interval of time. The cooling period
is characterised by larger peaks of energy with respect to the strongly stratified one, reaching
maximum values in November. A trend towards a longer period is also detectable. Conversely, the
deeper oscillations (Fig. 3d) show higher energy content but appears much more dispersed between



1 day and 4 days periods. Three periods with distinctive features may be identified: the first two
months (Aug-Sept) are characterized by the stronger thermal stratification and four major peaks are
detectable in the 2-4 days band; in the following two months (Oct-Nov), during the autumn cooling,
the energy level is lower and is centred around 1 day; in the final two months (Dec-Jan), when the
thermal stratification is weak, the peaks of energy are maximum and are sparse in the 2-4 days band.
In order to better highlight these different behaviours, Figures 4-6 show an analysis of a representative
fraction of each of these periods. Figure 4 refers to the third week of October where both the
oscillatory motions at the depth of the thermocline and the oxycline have energy peaks centred around
a daily period (see Figure 3). During this week, the wind speed and direction show the typical pattern
of the prealpine lakes (Valerio et al., 2017), blowing regularly southerly during the day and northerly
during the night. The upper water body is thermally stratified, with a thermocline at 18 m of depth
that separates a well-mixed epilimnion (16.7 °C) to the metlamnion below. The internal wave
response in the upper 30 m is a regular daily motion with amplitude around 6 m, clearly detectable
by the vertical fluctuations of the 15 °C isotherm highlighted in Figure 4c. Deeper in the water, the
main vertical fluctuations at LS-S shows a similar response both in term of amplitude and periodicity,
even though less regular and superimposed to higher frequency signals. At 24:00, a downwelling
event in the epilimnion at LS-N corresponds to a respective upwelling around the oxycline at LS-S,
while at 12:00 an upwelling at LS-N corresponds to a downwelling at LS-S. Accordingly, the 0.5 mg
$L^{-1}$ iso-oxygen at 95 m of depth at LS-S is dominated by a 1 day period wave oscillating in
counterphase with respect to the 15 °C isotherm at the other end of the lake, suggesting a H1V1
structure behaviour. Consistently with a H1 structure, the DO signals measured at 95 m at LS-N and
at LS-S are oscillating in counterphase (see Fig. 4e).
On 21-22 July 2017, when a similar situation dominated by the V1H1 mode was present, we collected
several vertical profiles to analyse the vertical structure of this motion. Figure 7 compares the
temporal evolution at LS-N and LS-S of the vertical profiles of temperature and oxygen around the
thermocline and the oxycline, respectively. One can easily see that the downwelling of the epilimnetic
waters at LS-N is also present in the deeper part, even though a bit vertically amplified and
characterized with a more irregular behaviour. At the same time, upwelling at all the depths occurs at
LS-S. These movements lead to simultaneous changes in redox conditions of the waters between a
depth of 100 m and 110 m.
A completely different oscillatory response is shown in Figure 5, which is referred to the period 28/8-
7/09 2017, when the continuous wavelet transform highlights a different frequency content of the
upper and deeper motions (see Fig. 3). Consistently, the contour of temperature and oxygen at the



different depths appear to be decoupled (see Fig. 5c-d). The water column is strongly stratified and
well described by a three layers structure with interfaces at -15 m and -35 m of depth. In the upper 35
m (see Fig. 5c), the 18°C and 12°C isotherms depths at LS-N, calculated by linear interpolation from
the temperature data, show a dominant 1 day oscillation in response to the daily alternation of the
southerly and northerly wind. Previous studies (Valerio et al. 2012) already interpreted this motion
as a V1H1 mode. A spectral analysis of the energy content of the metalimnetic thickness suggests the
superimposition of a lower amplitude, longer period (about 3 days), second vertical (V2) mode
oscillation. This is qualitatively evident in Figure 5c after the strong northerly wind started on the 1[th]
of September, which induced a distinctive metalimnion stretching on the 2[nd] and metalimnion
squeezing on the 3[rd]. The relative importance of these two different modes may be better quantified
by low-pass filtering the 12°C oscillations with different cut-off periods. To this end, we made use of
a Butterworth filter of the 4[th] order, imposing a range of periods around the peaks highlighted in the
spectrum. A ~ 1 day oscillation with an average 3.3 m amplitude captures the main oscillation pattern
(RMSE = 0.99 m), but the superimposition of an oscillations with an average 2.2 m amplitude and
period between 2 and 4 days allows to improve the fit (RMSE = 0.76 m), by better capturing the
downwelling of the epilimnion on the 29[th] of August and on the 2[nd] and 4[th] of September. Deeper in
the water (see Fig. 5d), the internal wave field is strongly different, as clearly highlighted by the much
larger excursions of the oxycline (up to 20 m) and their longer periodicity. At 85 m, this motion gives
rise to alternating oxygen concentrations with values ranging from zero (0.1-0.3 mg L$^{-1}$) up to 2.2 mg
L$^{-1}$ (see Fig. 5e). In order to identify more quantitatively the dominant signals at this depth, we
repeated the analysis developed before, this time using a low-pass filtering of the 0.5 mg L$^{-1}$ iso-
oxygen line at LS-S. In this case, the main oscillation pattern is described by filtering the signal with
2 and 4 days period and average 14.6 m amplitude (RMSE = 3.0 m), while the superimposition of a
daily oscillation with an average 2.2 m amplitude allows to further improve the fit (RMSE = 2.4 m).
With regard to the spatial structure of the observed lower frequency motion, the comparison of the
data obtained from the different stations suggests a V2H1 mode. Figure 5e shows that the ~ 3 days
oscillation of the DO immediately above the oxycline is in counterphase at LS-N and LS-S. Moreover,
one can observe the typical pattern of an higher vertical mode: the metalimnion stretching at LS-N
(e.g. 2 September), which is associated with a downwelling in the hypolimnion, occurs synchronously
with a hypolimnetic upwelling at LS-S. Accordingly, in the period under consideration, the field data
suggest the dominance of a V1H1 mode in the epilimnion and of a V2H1 mode in the hypolimnion.
On 16-18 April 2018, when a similar situation dominated by V2H1 was observed in the hypolimnion,
we measured vertical profiles to examine the vertical structure of this motion at higher resolution.
The DO time series measured at LS-N and LS-S at the oxycline depth (see Fig. 8a) shows a distinctive



H1 oscillation with a period around 4 days. In correspondence to the maximum and minimum vertical excursion of this fluctuation, we compared the vertical profiles at LS-N of temperature and oxygen around the thermocline and the oxycline, respectively. Contrary to what was observed in Figure 7, the oscillation is not vertically uniform: a downwelling of the thermocline in the order of a few meters is associated with an upwelling of the oxycline of about 25 m. Accordingly, from these data further highlight the amplification of the vertical excursion of the V2H1 motion in the area of the oxycline.

An intermediate response with respect to the previous ones is that shown in Figure 6. In this case, referred to winter 2017, the water column is weakly thermally stratified and characterized by a 9.8 °C epilimnion between 0 to 35 m of depth. The wind is mostly southerly (see Fig. 6b) and the wavelet transform of the DO measurements at LS-S show a first tight peak around a 4 day$^{-1}$ frequency and a second around 2 days$^{-1}$ (see Fig. 3d). Consistently, the time series of the 0.3 mg DO L$^{-1}$ measured at LS-S (see Fig. 6d) shows evidence of both a shorter and a longer period signal. Filtering with the cut-off periods highlighted in the spectrum show that these ~2 and ~4 days signals have in this case comparable amplitudes (5.0 and 7.6 m, respectively) around the chemocline at LS-S. At LS-N, the time series of DO at 85 m oscillates in counter phase with respect to the southern one with coherent periodicities and amplitudes, so suggesting a H1 response (see Fig. 6d). With regard to the vertical structure, the only measurements available for a comparison are the high-resolution temperature measurements between 55 and 105 m (see Fig. 6c), because the thermistors above do not measure with enough accuracy to detect the small temperature variations in this period of the year. At these depths, no clear evidence of squeezing or stretching of the intermediate layer is detectable. However, conversely to what observed at LS-S, the ~2 days signal is dominant over the ~4 days. This might indicate an attenuation of the longer period signal at lower depths, which characterizes the V2 motions.

### 3.2 Analysis of the model results

With reference to the results of the modal model, Table 2 reports the monthly-averaged values used for the calculations, and the obtained yearly evolution of the periods of V1H1, V2H1 and V3H1 mode. In April, all the periods present large values (V1H1: 2.9, V2H1: 4.6 and V3H1: 6.4 days) that progressively decrease during the warming season. In the strongly stratified period (July-October) the modelled periods of V1H1, V2H1 and V3H1 show almost constant values, around 1, 3 and 4 days, respectively. As soon as the water column starts to cool and the thermocline deepens, all mode periods start growing. During the weaker 3-layers stratification in February, when a similar density difference



is present across the metalimnion and the chemocline, V2H1 reaches a 7.4 days period, while the
V1H1's period increases up to 3.3 days in March, when the water column is thermally homogeneous
and only a salinity stratification is present.
With regard to the spatial structure of these modes, Table 2 summarizes the 3D results in terms of
maximum interface displacements at different lake locations in four representative periods of the year.
In the following, we will mostly focus on the V1H1 and V2H1 oscillations of the thermocline ($\xi_2$) and
the chemocline or oxycline ($\xi_4$) at LS-S and LS-N to provide an interpretation of the data measured at
these stations from July 2017 to February 2018. With regard to the first vertical mode, all the
interfaces oscillate in phase at the different depths. At LS-N (220 m deep) their amplitude are almost
vertically uniform ($\xi_4/\xi_2 \approx 1$), while at LS-S and NB (105 m deep) the intermediate and deep tilt is
amplified ($1.2 < \xi_4/\xi_2 < 2.3$). Conversely, the deep interface's tilt is strongly damped and more irregular
in the eastern basin (EB), a 100 m flat plateau located east of Monte Isola ($\xi_4/\xi_2 \approx 0.4$). In absolute
terms, the weaker is the density stratification, the larger are the interface tilts. At the end of the winter,
when the water column in thermally homogenous and chemically stratified, the V1H1 amplitudes
become up to 7.5 times larger than the summer thermocline's one. For example, at LS-N under the
action of the synthetic wind favouring a V1H1 mode, the upper interface oscillates in August with a
5.3 m amplitude around 15 m, while in March it oscillates with a 20.1 m amplitude around 95 m.
With regard to the second vertical mode, the interfaces oscillation is strongly non-uniform over the
vertical, with the metalimnion and the chemocline both oscillating in counterphase with respect to
the upper thermocline and with much larger vertical displacements. At LS-N, the vertical
displacement of the chemocline is on average 2.6 times larger than the thermocline's one. This
vertical amplification is favoured by larger density gradients (at LS-N $\xi_4/\xi_2$ decreases from 3.4 in
August to 1.8 in December). Similarly to what was observed for V1H1, this vertical amplification is
also enhanced at the southern end of the lake (average $\xi_4/\xi_2 = 4.0$ at LS-S), while it is strongly
attenuated in the eastern basin, where the chemocline oscillations are more irregular in time and show
maximum vertical displacements comparable to the thermocline's ones ($\xi_3/\xi_1 \approx 1$). For the discussion
that will follow, it is worthy to underline that, independently from the vertical mode and the
stratification, the ratio between the V1H1 and V2H1 amplitude of a given interface simulated at
different lake locations does not present a wide range of variation. In particular, the displacement of
the deeper interface $\xi_4$ at the different location is: $0.6 < \xi_{4\text{-LS-N}}/\xi_{4\text{-LS-S}} < 0.8$; $0.1 < \xi_{4\text{-EB}}/\xi_{4\text{-LS-S}} < 0.2$; $0.7 <$
$\xi_{4\text{-NB}}/\xi_{4\text{-LS-S}} < 1.2$. This implies that the chemocline keeps a similar H1 horizontal structure throughout
the year, even though with different absolute values depending on the stratification and the vertical
mode. This comes clearly to light looking at the oxygen distribution at 95 m of depth simulated in
correspondence of the maximum tilt of the chemocline for a V1H1 and V2H1 mode (see Fig. 9). To



clarify the reason for the limited contribution of the eastern basin (EB) to the oscillation of the
chemocline, we conducted some simulations by modifying the bathymetry of the lake. Actually, the
proximity to the central part of the basin explain only a limited fraction of the amplitude's reduction,
which resulted instead mostly due to the lake's bathymetry. If EB would be as deep as the central
basin (z = 250 m), we estimated that the chemocline tilt at EB $\xi_{4\text{-LS-EB}}$ would be on average three times
larger than the actual one, and would be about 50% of the one simulated ad LS-S ($\xi_{4\text{-LS-EB}}$/ $\xi_{4\text{-LS-S}} \sim 0.5$).
The analysis of the measured data previously shown suggests the presence of both a V1H1 and a
V2H1 response. The obtained numerical results allowed us to clarify the nature of these oscillations
and extend the spatial information provided by local measurements.
The field data showed that the thermocline oscillates regularly over a 1 day period from July to
November (Fig. 3b). At that time, the natural period of V1H1 is daily, too, confirming that the upper
water layers are dominated by this type of motion. Conversely, for the deeper oscillations we observed
a different type of response: a shorter daily oscillation in Oct-Nov (e.g. Fig. 4), and a longer 2-4 days
oscillation from August to September, and from December to January (e.g. Fig. 5, 6). In the first case,
(Oct-Nov), the daily period of the measured oscillations fits the natural period of V1H1. This is
consistent with the spatial structure of this motion (see e.g. Fig 4 and related comments), characterized
by a counter-phase response at the two lakes ends (H1) and a similar amplitude at the different depths
(V1). In the strongly stratified period (Aug-Sept), we occasionally observed a decoupled internal
wave response at the different depths. By comparing the periodicity of the measured oscillations and
that of the natural modes (see Fig. 3), the thermocline appears to be dominated by a V1H1 motion (~
1 day period), while the oxycline by a V2H1 motion (~ 2-3 days period). Again, this is consistent
with the observed spatial structure of these motions (see e.g. Fig. 5 and related comments). It is of
major interest to observe that in correspondence of the summer stratification at LS-S, the deeper
amplitude $\xi_4$ of the V2H1 mode results 5.7 times larger than $\xi_2$ . This is likely to explain why V2H1
mode is largely dominant deeper in the waters, while it is sheltered by the V1H1 oscillations around
the thermocline. Finally, in the third period (Dec-Jan) we observed the superimposition of ~2 days
and ~4 days large oscillations at the depth of the oxycline. According to the periodicities of the natural
modes, they correspond to a V1H1 and V2H1 mode (see Fig. 3e). With respect to the previous case
(Aug. 17), the evidence of a large amplitude V1H1 mode at this depth is consistent with the increased
displacements reported in Table 3 in correspondence of a weaker density stratification.





In conclusion, this analysis provides an interpretation of the physical nature of the oscillations of the
oxycline observed from July 2017 to February 2018 in the southern part of Lake Iseo. At this point,
it is of interest to reflect upon the reasons of the excitations of these motions. In the upper waters,
Valerio et al. (2012) showed that the internal wave modes are excited whenever the spatial and
temporal structure of a wind field over a lake matches the surface velocity field of a particular internal
mode. Along the same line of thought, in Figure 3e we superimposed the natural frequencies of the
two main vertical modes to the continuous wavelet transform of the northerly components of the wind
forcing, to see whether a fit in their periodicies might explain the observed internal wave motions.
During the stratified period (July – November), most of the wind energy oscillates in a period of
around 1 day, likely due to the regular alternation of northerly and southerly thermal winds, typical
of this area (see Valerio et al. 2017). This forcing perfectly fits the V1H1 mode that is regularly
excited and dominates the response of the upper waters (see Fig. 4b). Occasionally, the daily wind
energy reduces its intensity when the wind blows longer from the same direction. From July to
October 2017, this happened three times (see Fig. 3e). Interestingly, in correspondence of each of
these events there is a large peak of energy in the oxycline oscillations for lower frequencies (see Fig.
3e). The reason lies in a resonance condition between the wind and the waves: the longer periodicity
of the wind forcing approaches the natural periodicity of the V2H1, so that it is excited in place of
V1H1, inducing large vertical fluctuations below the metalimnion. During the weakly stratified
period, the thermal conditions in the surrounding watershed limit the intensity and the regularity of
the alternated thermal winds, causing the spreading of the wind energy over a larger band of
frequencies (see Fig. 3e). Contrary to what happened before, this condition favours the excitation of
a longer period V1H1 oscillation, that then clearly appears also at the depth of the oxycline with
amplitudes favoured by the weak stratification.

## 4. Discussion and Conclusions

In Lake Iseo the reduced deep mixing has determined the formation of an anoxic monimolimnion
below 95 m, so that any vertical displacement of the oxycline may induce variation in the redox
conditions of the contiguous sediments. Accordingly, it seems reasonable to advance the hypothesis
that the internal wave motions in Lake Iseo might force unsteady sediment–water fluxes.
The data collected from July 2017 to February 2018 clearly substantiate our initial hypothesis that
there are large and periodic displacements of the oxycline. The oxycline typically oscillation in the
southern basin is in the range 10 – 20 meters, with periods ranging from 1 to 4 days. Comparing these
movements to those already studied in the lake's upper water layers (Valerio et al. 2012), dominated



to a large extent by a 1 day motion, we found the dynamics in deeper waters to be of more irregular
character, featuring larger amplitudes at lower frequencies. We attributed this behaviour primarily to
the excitation of a first horizontal, second vertical (V2H1) mode, which is characterized by amplified
vertical displacements below the thermocline. During weakly stratified conditions, instead, this
behaviour was also explained by the excitation of a first horizontal and first vertical (V1H1) mode,
featuring lower frequencies and larger tilts due to the weaker density gradients. In both cases, the
overlap of the temporal structure of the wind forcing with that of the two modes provides evidence
for their excitation by wind, suggesting a resonant response to wind as observed, inter alia, by Vidal
et al. 2007.
A primary role for the excitation of these deep internal waves motions is played by the presence of a
permanent chemical stratification. In Lake Iseo, a depth variation of the mineralization process along
the water column generates a gravity driven segregation with a density gradient between the oxic
mixolimnion and the anoxic monimolimnion, which favours the occurrence of large baroclinic
motions at the interface of these layers, even if the water column is thermally homogenous.
Accordingly, this works provide experimental and numerical evidence of a chemical gradient
supporting deep baroclinic motions in perennially stratified lakes, as already argued numerically by
Salvadé et al. (1988) in Lake Lugano, where a similar density stratification is present. Being the
chemocline at the same depth of the oxycline, the oscillations of the former induces alternating redox
conditions in the water above the sediments.
The observations of highly energetic V2H1 motions in lake Iseo widen the observations of higher
vertical modes, which have been much less frequently reported in lakes with respect to the first
vertical mode (e.g. Wiegand and Chamberlain, 1987; Münnich et al. 1992, Roget et al., 1997), and
rarely reported in deep lakes (e.g. Boehrer et al. 2000; Guyennon et al. 2014). Interestingly, the
stronger evidence of these motions in the deep waters with respect to the upper ones endorses the idea
by Hutter et al. (2011) that the reason for the rare documentation in the literature of these motions is
not due to the fact that they are not excited as much as to the measuring techniques that are usually
not been sufficiently detailed to capture them. Moreover, if typically their excitation was seen to be
favoured by the presence of a thick metalimnion, in this case they are enhanced by the presence of a
chemical stratification in the deeper waters. By conducting a sensitivity analysis on the density
stratification used as initial condition for the simulations, we observed e.g. that in August that the
amplitude of  the second vertical modes is from 2 to 3 times larger with respect to a case where the
chemical stratification would be absent. Interestingly, similar observations of higher baroclinicity
supported by deep isopycnal layers were observed in the marine environment in correspondence of





vertical salinity gradients (South Aral Sea by Roget et al. 2017), and turbidity (Ebro Delta by Bastida
et al., 2012).
The observation of a pronounced strong spatio-temporal dynamics of the oxycline strongly supports
the hypothesis of the occurrence of alternating redox conditions in a large portion of the bottom of
Lake Iseo. This may be intuitively understood by observing the DO value e.g. at 95 m of depth in the
(d) panels of Figures 4 to 6. The measured variations of DO imply that the surface sediment surface
at this depth is likely to be subjected to a variation of DO concentrations in the overlying water of
between 0 and 3 mgL$^{-1}$. To quantify the potential biogeochemical implications of the oxycline
dynamics in Lake Iseo, it is important to estimate the extent of sediment area subjected to alternating
redox conditions. Obviously, the smaller the slope, the larger will be the sediment area impacted by
a given vertical displacement of the oxycline. Accordingly, the areas subjected to alternating redox
conditions will be mainly located in the northern, southern and eastern sub-basins (see Fig. 10a),
where the bottom rises more gradually compared to the bathymetry of the steep central basin. The
depth-area relationship of the three sub-basins (see Fig. 10a) shows that approximately 5 km$^2$ of
sediment area are situated between the depths of 85 and 115 m, representing an upper bound of the
area subjected to episodic changes in oxygen availability. However, for a more precise calculation, it
is necessary to account for the actual oxycline oscillations in the three sub-basins. To this end, we
used the time series of vertical displacements of the 0.5 mgDOL$^{-1}$ iso-oxygen line at LS-S as a
reference for the oxycline depth in the southern basin. When associated to the area-depth curve of the
southern basin, these data allowed a computation of the time series of the anoxic area in that sub-
basin. The analysis of its oscillations over a 3 days window provided the time series of the area
subjected to variable redox conditions (see the blue area in Fig. 10b). The same analysis was extended
also to the northern and the eastern basin. To estimate the oxycline oscillations there, we accounted
for the spatial structure of the deeper layer interface provided by the 3D numerical model (see Tab.
3), which keeps a similar first horizontal H1 structure throughout the year. By comparing the
amplitudes in different points of the lake, we could use the time series of vertical displacements of
the 0.5 mgDO/l at LS-S to extrapolate an analogous time series for the northern and the eastern basin.
The resulting basin-specific areas subjected to alternating redox conditions are shown in Figure 10b.
We estimated that overall 1.9 km$^2$ of bottom sediments of Lake Iseo, 3% of the whole area, are on
average subjected to alternating redox conditions with periods from 1 to 4 days. Local maximum
areas are reached in summer, when long-lasting winds favour the excitation of a second vertical mode,
and after December, when the weak thermal gradients favour strong tilts of the chemocline. The
relative contributions of the sub-basins to the whole area subjected to alternating redox conditions are



49% (southern basin), 32% (northern basin) and 19% (eastern basin). Interestingly, a non-negligible
relative contribution comes from the eastern basin, even though it is characterized by a strong
attenuation of the deep internal wave dynamics. In this area, the oscillations have on average 90%
lower amplitudes with respect to the southern basin. The reasons lie in the small slope of the area-
depth curve between 95 and 110 m, where oscillations with an amplitude of 5 m impact roughly 2
$km^2$ of sediment area. Though, given the observed irregularity of the signal simulated at EB, further
experimentally verification of the oxygen variations at the bottom of the eastern basin seem
warranted.  Finally, it is important to stress that the EB contribution is present only if the oxycline
oscillates around 100 m. In case it would be located markedly above or below, only the northern and
the southern basin would be affected by alternating oxygen availability.
A dynamic oxycline, covering a substantial fraction of a lake's sediment area can have implications
for the lake-internal redox processes and several previous studies suggested that seiche-driven oxygen
fluctuations have effects on benthic biogeochemical turnover. Implications are further expected for
the redox-sensitive sediment phosphorus release (Søndergaard, 2003), however, the P release upon
exposure to anoxic excursions was found to be more subtle than expected, with 95% of P remaining
particle-bound independent from $O_2$ oscillations (Parsons, 2017). In order to concur with the current
understanding of benthic nutrient cycling, these results suggest that redox-sensitive Fe and Mn
(oxyhydroxides) in the sediment may indeed release surface-bound P in oxygen-depleted conditions
(Søndergaard, 2003), which can, however, to a large extent be redistributed within the other sediment
fractions (Hupfer and Lewandowski, 2008; Parsons et al., 2017). Accordingly, it is not surprising that
the P binding potential of sediments that are regularly in oxic but now temporarily exposed to anoxic
waters are not found to differ from those in permanently oxic conditions (Aller, 1994). In contrast to
that, the role of sediments as sink and source of P in regularly anoxic environments is known to be
controlled by other diagenetic processes including P supply, microbial mineralization, aluminium and
sulphide availability (Hupfer and Lewandowski, 2008). However, the collective susceptibility of
these processes to excursion in oxygen availability is less well understood.
As a result, the reported mechanisms of short-term, wind-induced fluctuations of the deep oxycline
implies that an additional 3% of the sediment area of Lake Iseo features the mixolimnetic P retention
in the current conditions of the lake. The deep waters of lake Iseo store the vast majority of the in-
lake P (360t of 480t, April 2016), indicating the relevance of P release from sediment in permanently
anoxic condition (Lau et al., in preparation). Therefore, it remains crucial to further explore the
dynamics in redox forcing on the sediments of perennially stratified lakes and the entailing
implications for internal P cycling and biogeochemical turnover.



**Competing interests**
The authors declare that they have no conflict of interest.

**Acknowledgments**
This research is part of the ISEO (Improving the lake Status from Eutrophy to Oligotrophy) project
and was made possible by a CARIPLO Foundation grant number 2015-0241.



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





**Table 1**. Summary of the oxygen data measured in Lake Iseo at a sampling frequency of 1 min$^{-1}$.







| ID | Station | Depth (m) | Distance from the bottom (m) | Investigated period |
|---|---|---|---|---|
| SO85 | LS-S | 85 | 20 | 21/07/2017 – 18/04/2018 |
| SO90 | LS-S | 90 | 15 | 21/07/2017 – 18/04/2018 |
| SO95 | LS-S | 95 | 10 | 21/07/2017 – 18/04/2018 |
| SO105 | LS-S | 105 | 0 | 21/07/2017 – 18/04/2018 |
| NO85 | LS-N | 85 | 135 | 24/10/2017 – 16/04/2018 |
| NO95 | LS-N | 95 | 127 | 22/07/2017 – 24/10/2017 |






**Table 2**. Progression of the natural periodicity of the first horizontal, first, second and third vertical modes in Lake Iseo over a one year period. The monthly-averaged layered structure used for the calculation is specified by $Z_i$, the elevation of the upper interface of each $i^{th}$ layer, and $\rho_i$ its density, expressed as a deviation from 1000 kgm$^{-3}$.

| Time | Layered structure | | | | | | | Periods of the H1 modes | | |
| --- | --- | --- | --- | --- | --- | --- | --- | --- | --- | --- |
| | $Z_2$ | $Z_3$ | $Z_4$ | $\rho_1$ | $\rho_2$ | $\rho_3$ | $\rho_4$ | V1 | V2 | V3 |
| | (m) | | | 1000-(kgm$^{-3}$) | | | | (hours) | | |
| Jul-17 | 12.5 | 35.0 | 95.0 | 1.764 | 0.287 | 0.093 | 0.064 | 26.7 | 65.1 | 88.9 |
| Aug-17 | 15.0 | 35.0 | 95.0 | 1.805 | 0.268 | 0.094 | 0.064 | 24.1 | 60.3 | 90.3 |
| Sep-17 | 17.5 | 35.0 | 95.0 | 1.550 | 0.251 | 0.095 | 0.064 | 23.7 | 65.4 | 92.5 |
| Oct-17 | 20.0 | 35.0 | 95.0 | 1.113 | 0.256 | 0.096 | 0.064 | 25.9 | 69.2 | 94.1 |
| Nov-17 | 22.5 | 35.0 | 95.0 | 0.710 | 0.252 | 0.099 | 0.065 | 31.0 | 74.4 | 102.8 |
| Dec-17 | 35.0 | - | 95.0 | 0.279 | 0.097 | - | 0.065 | 43.8 | 82.3 | - |
| Jan-18 | 45.0 | - | 95.0 | 0.125 | 0.094 | - | 0.066 | 67.7 | 139.2 | - |
| Feb-18 | 55.5 | - | 95.0 | 0.112 | 0.093 | - | 0.066 | 71.2 | 177.3 | - |
| Mar-18 | - | - | 95.0 | 0.091 | - | - | 0.066 | 78.5 | - | - |
| Apr-18 | 7.5 | 35.0 | 95.0 | 0.260 | 0.131 | 0.087 | 0.060 | 69.3 | 112.3 | 153.5 |
| May-18 | 10.0 | 35.0 | 95.0 | 0.656 | 0.167 | 0.088 | 0.060 | 48.4 | 75.8 | 108.7 |
| June-18 | 12.5 | 35.0 | 95.0 | 1.133 | 0.192 | 0.089 | 0.061 | 33.6 | 69.1 | 101.3 |





**Table 3**. Maximum vertical displacement ξ of the layer interfaces with respect to their equilibrium
level for the first three vertical modes in Lake Iseo at four different locations. These locations, whose
depth is z, are shown in Fig.1. The interfaces displacements were simulated with AEM3D by forcing
with a spatially uniform sinusoidal wind, with a maximum speed of 5 m/s and a period equal to the
natural one predicted by the modal model (see grey shading in Table 2). $\xi_i$ indicates the upper interface
of each $i^{th}$ layer, whose depth is reported in Tab. 2.

| | V1H1 | | | | | | | | | | | |
|---|---|---|---|---|---|---|---|---|---|---|---|---|
| | LS-N, z = 220 m | | | LS-S, z = 105 m | | | EB, z = 100 m | | | NB, z = 105 m | | |
| Time | $\xi_2$ | $\xi_3$ (m) | $\xi_4$ | $\xi_2$ | $\xi_3$ (m) | $\xi_4$ | $\xi_2$ | $\xi_3$ (m) | $\xi_4$ | $\xi_2$ | $\xi_3$ (m) | $\xi_4$ |
| Aug-17 | 2.7 | 2.4 | 2.9 | -2.2 | -4.5 | -5.1 | -1.1 | -1.1 | -0.6 | 2.6 | 4.1 | 5.3 |
| Oct-17 | 3.1 | 3.1 | 4.1 | -3.0 | -4.5 | -6.0 | -1.6 | -1.2 | -0.6 | 3.1 | 4.5 | 5.3 |
| Dec-17 | 6.1 | - | 7.2 | -7.7 | - | -9.2 | -2.9 | - | -1.0 | 7.5 | - | 10.9 |
| Mar-18 | - | - | 10.7 | - | - | -16.1 | - | - | -4.4 | - | - | 12.7 |

| | V2H1 | | | | | | | | | | | |
|---|---|---|---|---|---|---|---|---|---|---|---|---|
| | LS-N, z = 220 m | | | LS-S, z = 105 m | | | EB, z = 100 m | | | NB, z = 105 m | | |
| Time | $\xi_2$ | $\xi_3$ (m) | $\xi_4$ | $\xi_2$ | $\xi_2$ (m) | $\xi_2$ | $\xi_2$ | $\xi_3$ (m) | $\xi_4$ | $\xi_2$ | $\xi_3$ (m) | $\xi_4$ |
| Aug-17 | 1.8 | -3.3 | -6.2 | -1.6 | 3.8 | 9.1 | -1.2 | 1.6 | 1.4 | 2.0 | -5.0 | -6.0 |
| Oct-17 | 2.3 | -2.8 | -5.9 | -2.3 | 3.8 | 9.3 | -1.8 | 1.6 | 2.2 | 2.4 | -4.2 | -6.7 |
| Dec-17 | 4.0 | - | -7.3 | -4.8 | - | 11.2 | -2.5 | - | 1.4 | 5.2 | - | -8.8 |
| Mar-18 | - | - | - | - | - | - | - | - | - | - | - | - |

| | V3H1 | | | | | | | | | | | |
|---|---|---|---|---|---|---|---|---|---|---|---|---|
| | LS-N, z = 220 m | | | LS-S, z = 105 m | | | EB, z = 100 m | | | NB, z = 105 m | | |
| Time | $\xi_2$ | $\xi_3$ (m) | $\xi_4$ | $\xi_2$ | $\xi_3$ (m) | $\xi_4$ | $\xi_2$ | $\xi_3$ (m) | $\xi_4$ | $\xi_2$ | $\xi_3$ (m) | $\xi_4$ |
| Aug-17 | 2.2 | -3.0 | 4.4 | -1.8 | 3.4 | -6.1 | -1.4 | 1.9 | -1.3 | 2.5 | -4.2 | 6.3 |
| Oct-17 | 2.8 | -3.5 | 5.1 | -2.4 | 3.8 | -5.4 | -2.2 | 2.3 | -2.2 | 3.0 | -4.2 | 8.1 |
| Dec-17 | - | - | - | - | - | - | - | - | - | - | - | - |
| Mar-18 | - | - | - | - | - | - | - | - | - | - | - | - |






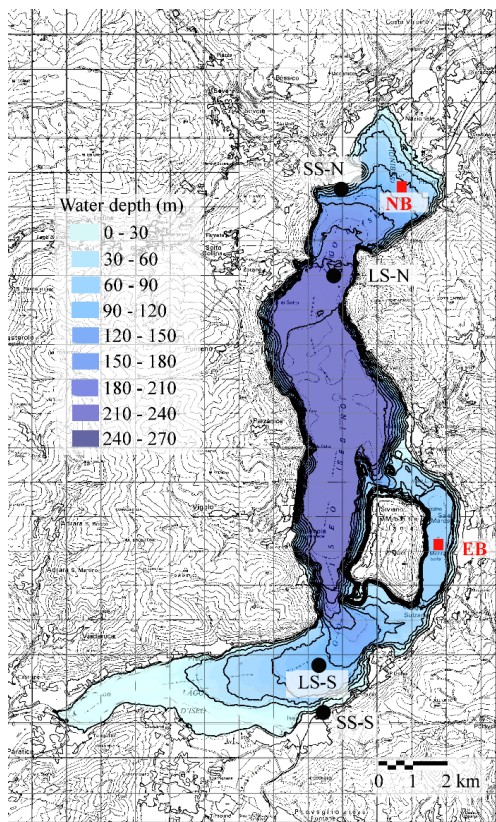

**Figure 1.** Bathymetry of Lake Iseo, represented with isodepth lines at 30-m spacing. The black dots show the measurement stations located on the shore (SS) and in the lake (LS), while the red squares indicate 2 points at 98 m in the eastern basin (EB) and 105 m in the northern basin (NB) that will be mentioned in the modeling section.




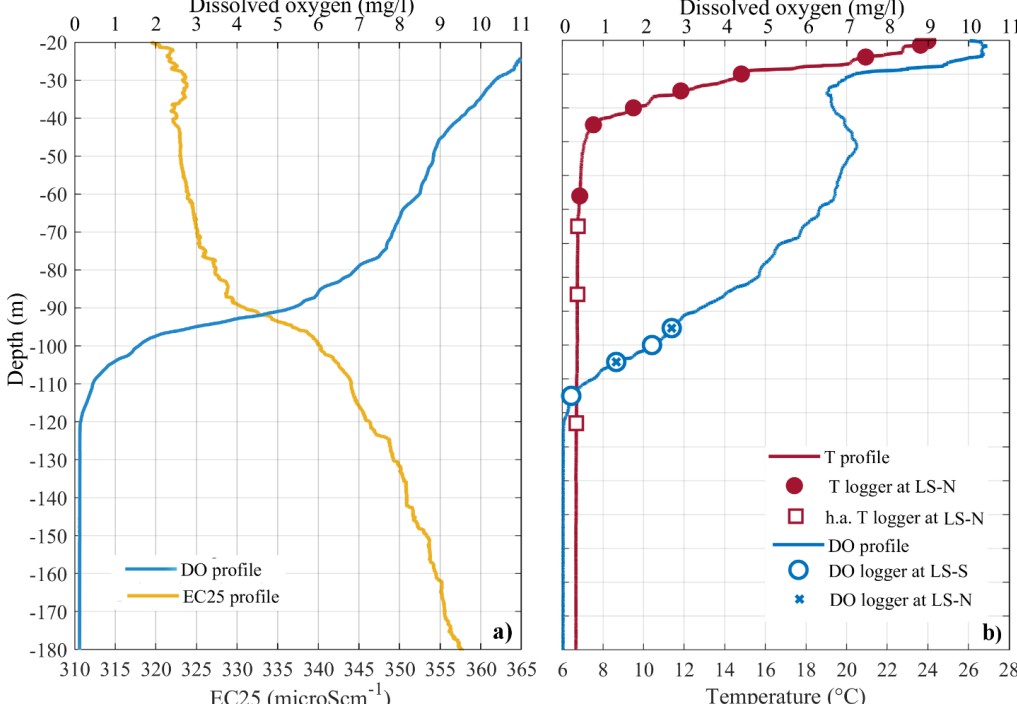

**Figure 2.** (a) Vertical profile of normalized conductivity (EC25) and dissolved oxygen (DO) measured on 10/04/2018 at the LS-N. (b) Vertical profile of temperature (T) and dissolved oxygen (DO) measured on 22/07/2017 at the LS-N. The circles and the crosses show the location of the dissolved oxygen sensors at LS-S and LS-N, respectively. The dots and the squares show the location of the temperature sensors at LS-N, with the squares indicating the high accuracy sensors.





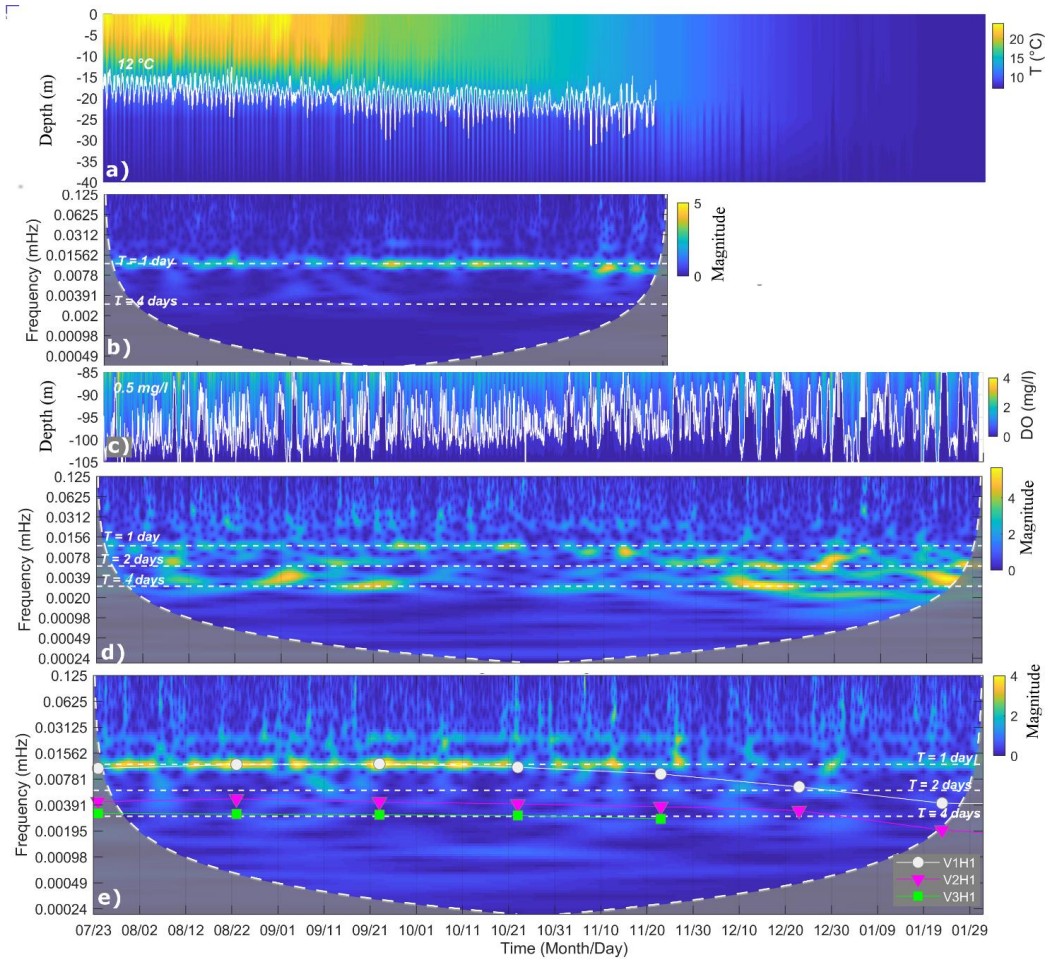

622

623

**Figure 3.** (a) Time series of the vertical displacements of the 12°C isotherm (white line) measured at LS-N, superimposed to the interpolated temperature distribution between 0 and 40 m, and (b) the associated continuous wavelet transform for the period between 23 July and 21 November 2017. (c) Time series of the vertical displacements of the 0.5 mgDO $L^{-1}$ isoline measured at LS-S, superimposed to the interpolated distribution of oxygen, and (d) the associated continuous wavelet transform. (e) Natural periods of V1H1, V2H1 and V3H1 mode superimposed to the continuous wavelet transform of the N-S component of the wind measured at LS-N. The grey shaded region on either end indicate the cone of influence, where edge effects become important.

632

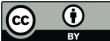



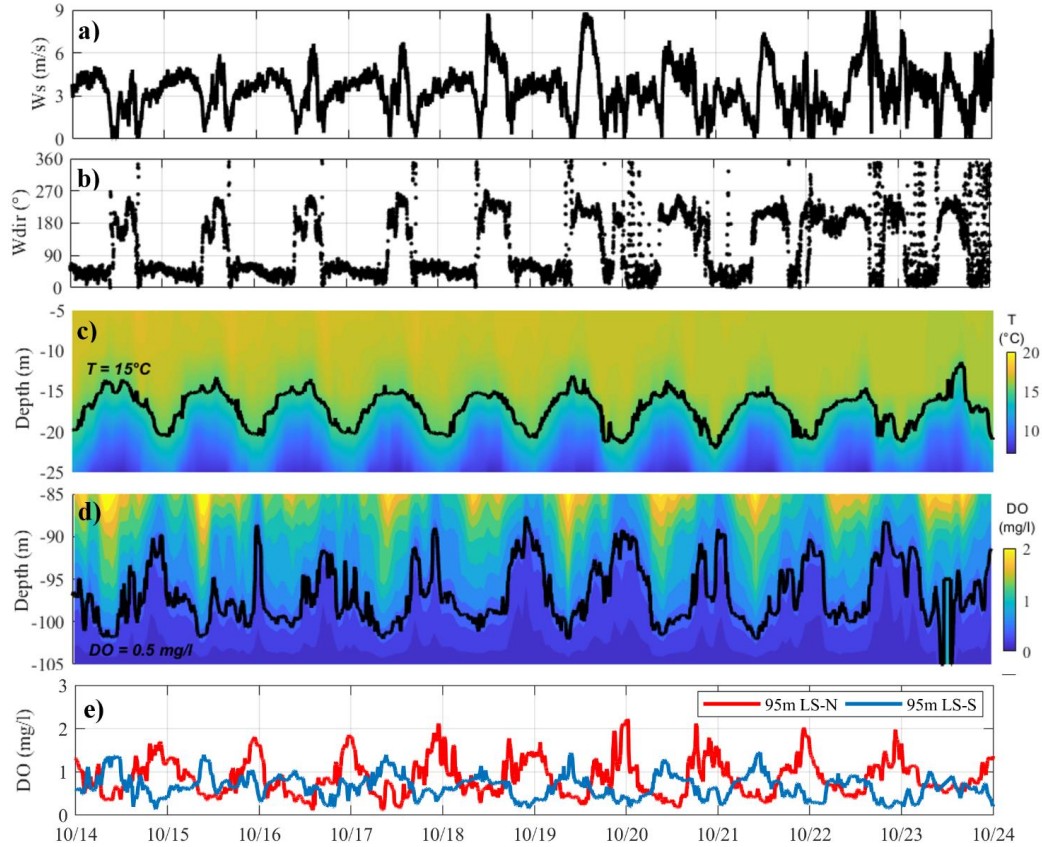

**Figure 4.** Time series measured from 14 to 24 October 2017 of (a) wind speed and (b) direction at
LS-N, followed by the spatial and temporal variation of (c) temperature between 5 and 25 m at LS-
N and of (d) dissolved oxygen between 80 and 105 m at LS-S. Panel (e) compares the time series of
DO measurements at LS-N and LS-S stations. In correspondence of each tick of the horizontal axis
it is 00:00 o'clock.





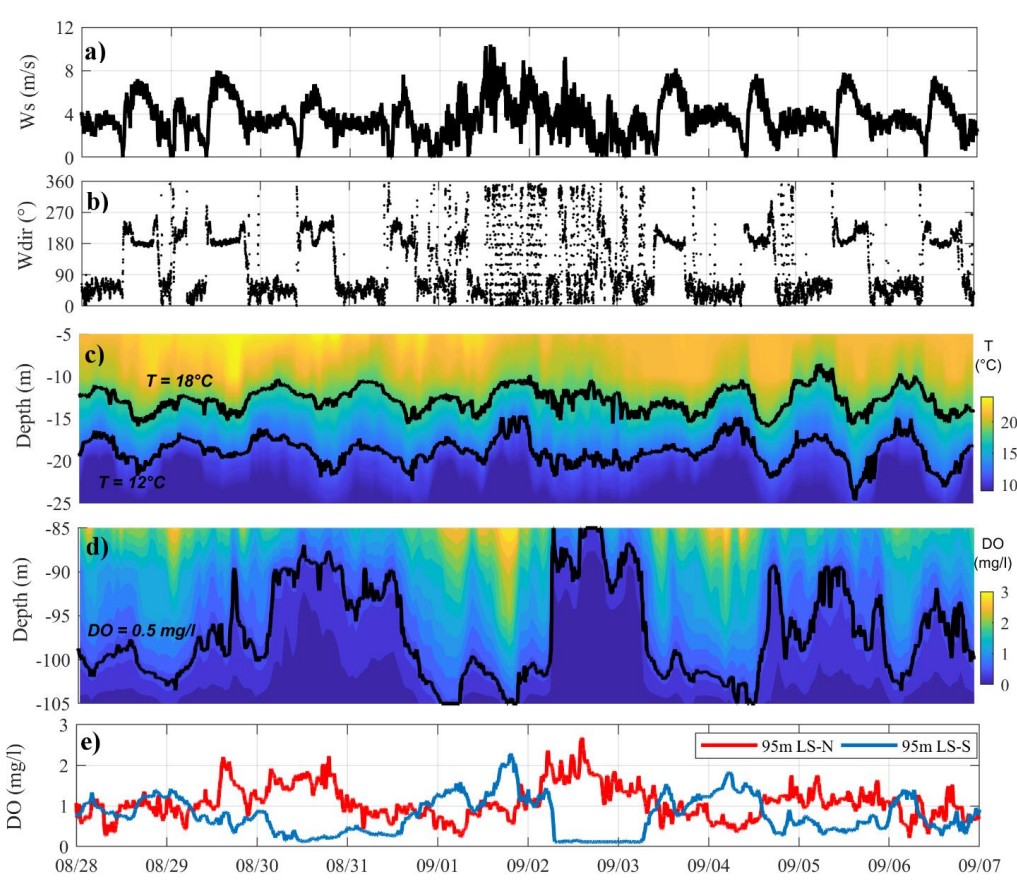



**Figure 5.** Same as Fig. 4 but referred to the period August – 07 September 2017.






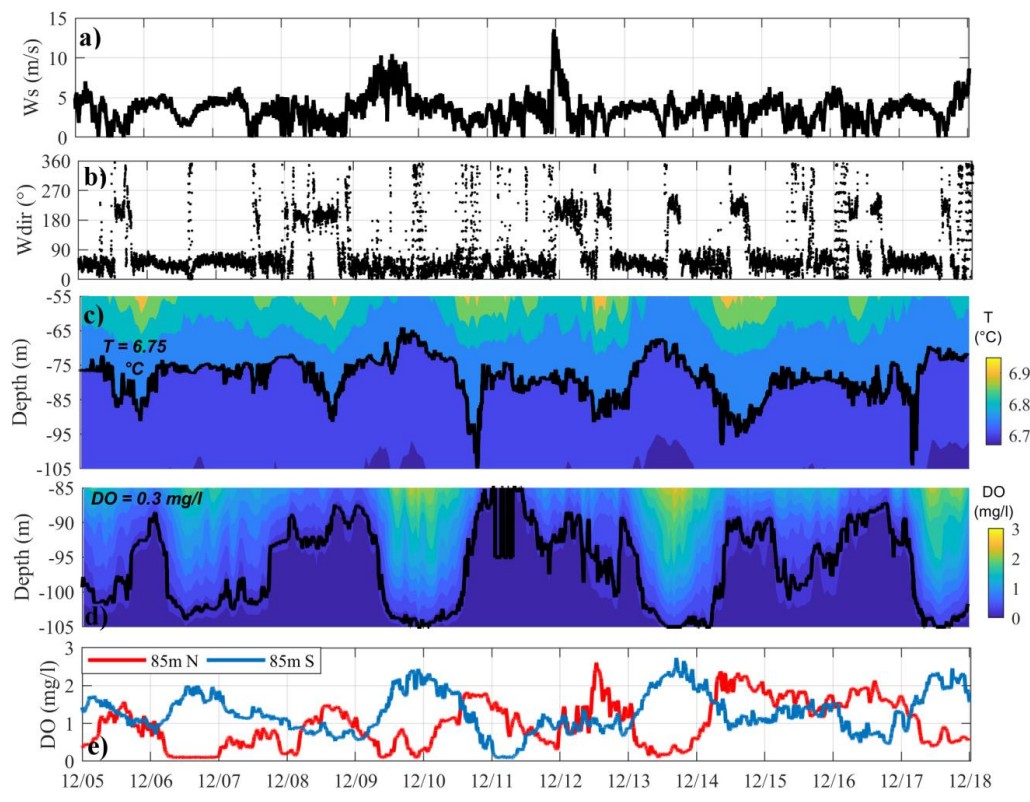



**Figure 6.** Same as Fig. 4 but referred to the period 05 – 18 December 2017.





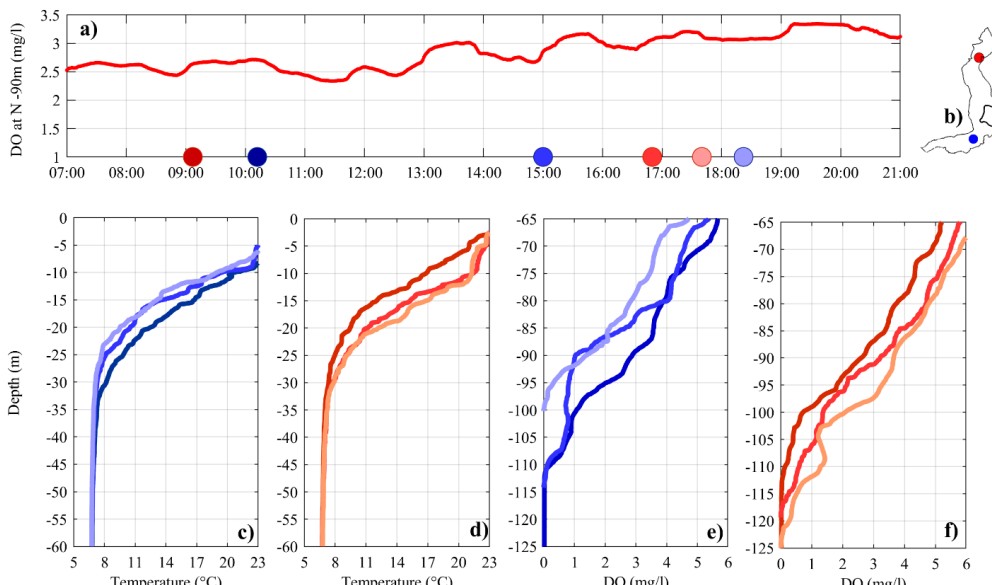



**Figure 7.** (a) Dissolved oxygen (DO) concentration at 90 m depth recorded at station LS-N on 21
July 2017. Coloured dots on the x-axis indicate the sampling time of the vertical profiles shown in
the panels (c-f). Panels (c) and (d) compare the profiles of temperature between 0 and 60 m, while
panels (e) and (f) compare the profiles of DO between 65 and 125 m. Red and blue colours refer to
the northern and southern sampling location (see panel b), respectively.


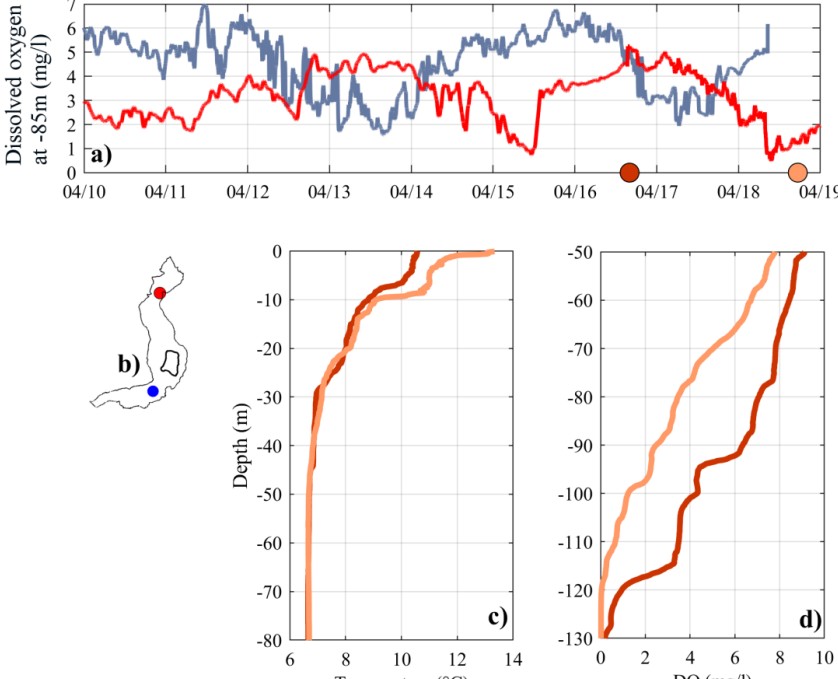

**Figure 8.** (a) Time series of DO measured at 85 m of depth on April 2018. Red and blue lines refer to the northern and southern sampling location, respectively (see panel b). Coloured dots on the x-axis indicate the sampling time of the vertical profiles shown in the panels (c-d) below. Panels (c) compares the profiles of temperature between 0 and 80 m at LS-N, while panels (d) compares the profiles of DO between 50 and 130 m at LS-N.




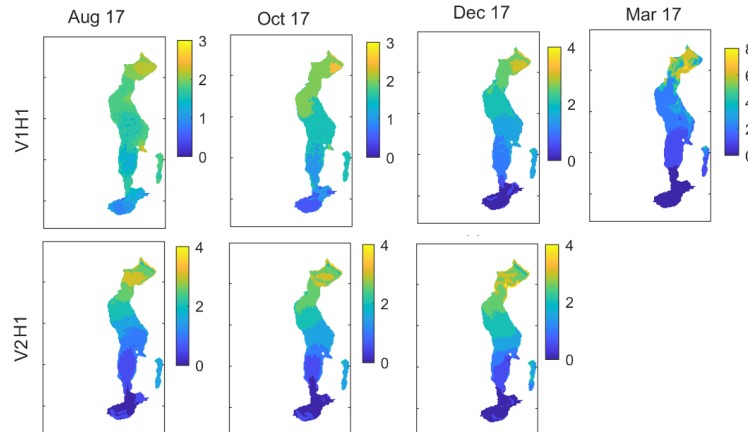



**Figure 9.** Contour of the oxygen distribution (mg L$^{-1}$) at 95 m of depth simulated with AEM3D in correspondence the maximum tilt of the oxycline. The panels refer to different vertical modes and different layered structures.






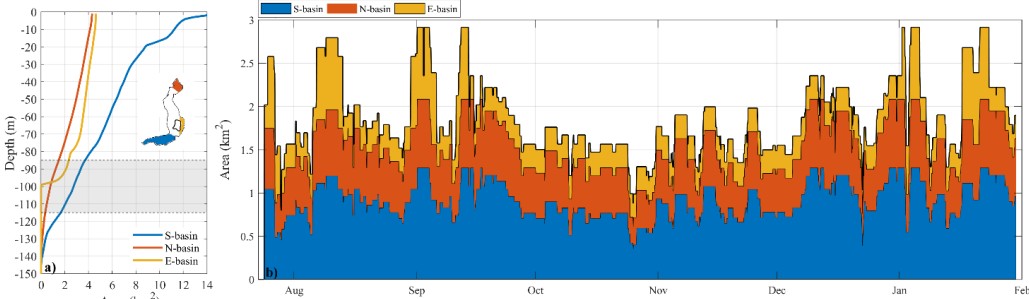

**Figure 10.** Estimation of the area of the bottom sediments subjected to alternating redox conditions. The areas were computed by considering a 3 days window. The three colours make reference to the contribution of the southern (S, blue), northern (N, red) and eastern basin (E, yellow), as shown on the map. In the left panel, the area-depth curves indicate the cumulative area of the bottom situated below a given water depth in each sub-basin. The grey shaded area marks the maximum and minimum vertical displacement of the 0.5 mgDO L$^{-1}$ recorded at LS-S, highlighting the area of the bottom where the oxycline fluctuates.