# Peer review of "Oxycline oscillations induced by internal waves in deep Lake Iseo"

_Hydrology and Earth System Sciences, 2018_

## Referee Comment (RC1) · Anonymous Referee #1 · 26 Nov 2018

review of HESS 2018-542 Oxycline oscillations induced by internal waves in deep Lake Iseo by Valerio et al.

This manuscript contains very interesting material from a well designed field programme and good displays of measurements that should warrant publication. However, the text at its current quality is not good enough for publication. I hope the authors consider my points and do a proper internal review before submitting an improved version. (Comments see below)

The manuscript deals with observations of internal waves in Lake Iseo and the monitoring of the oxycline motion over sediment at depths around 95m. The varying redox conditions impact on the geochemical processes at the sediment. The impacted area was estimated to 3% of the total lake bed. At times, the oxicline moved syncronously

with the thermocline V1H1 mode at one oscillation per day, but earlier in summer the oscillations of oxycline and thermocline were out of sync. The oxycline oscillated at 1 cycle in four days. Later in autumn, signals were not so clear but rouoghly the same fundamental oscillation period of close to 4 days can be seen at both depths.

The results are fascinating; these waves out of sync at various depth; after seeing Figs. 4,5,6, the reader is interested in hearing an explanation but the results section requires persistence to work one's way though the data processing details. In the end, I did not find an explanation for the a-syncronous behaviour.

comments on the content:

lines 271 and 272: should it be 1/4 * 1/day and 1/2* 1/day ?

Table 2: how have the depths, which separate the layers, been chosen? This choice has a major impact on the results.

No density profiles have been shown for the three selected periods and no comparison with the discrete layer structure has been shown.

Figs. 4,5,6 would the projection of wind on the lake axis not be more instructive than speed and direction?

Fig. 10 writing too small. add display of area vs depth for entire lake

Fig.2 caption Electrical conductivity is "temperature compensated" not "normalized"

Fig. 3 could indicate the periods covered in Figs 4,5,6

Table 3: line of V2H1 xsi indices are wrong

line 505: J. Ilmberger is NOT Jorg Imberger

For modes in lakes, the authors may also see these papers on Lake Constance: Boehrer 2000: modal response in a deep stratified lake: western Lake Constance Appt et al 2004: basin scale motion in upper Lake constance

comments on the text:

quite often the components of a sentence are not in the correct order: e.g. line 25 to just list one: "These basin-scale internal waves cause in the water layer between 85 and 105 m depth a fluctuation of the oxygen concentration between 0 and 3 mg L-1 that, due to the bathymetry of the lake, changes the redox condition at the sediment surface."

there are unnecessary words: as example: the first sentence of the paper line 33 "The physical processes occurring at ... " in stead of "Physical processes at ..."

inconsistent names in line 116: SS-1 and SS-2 refer to LS-N and LS-S on Fig.1 ?

line 297 wrong reference Table 2 should be Table 3

often narrative style: examples for the results section: line 319 "For the discussion that will follow, it is worthy to underline that, ... line 334 "The analysis of the measured data previously shown suggests the presence of ..." line 360 "At this point, it is of interest to reflect upon, ..."

difficult sentences: one example in lines 402-405: "In Lake Iseo, a depth variation of the mineralization process along the water column generates a gravity driven segregation with a density gradient between the oxic mixolimnion and the anoxic monimolimnion, which favours the occurrence of large baroclinic motions at the interface of these layers, even if the water column is thermally homogenous."

not optimal choice of words: line 167 " we thickened the grid ..." is there no better choice? line 235 "streching" ? is "widening" better? line 243 "strongly different"

line 468- 483 is this discussion material?

line 330 " the amplitude's reduction" ... "the lake's bathymetry"

line 144 "natural" modes? I know normal modes, or modes of the internal wave equation or Taylor - Goldstein eq.

The presented material has the potential for a good paper, but the writing needs to be improved. My recommendation is to get the results and the discussion focussed on the message of the paper. Shorten the text of these sections considerably. Make a proper internal review and if necessary ask for language editing.

———————————————

---

## Referee Comment (RC2) · Anonymous Referee #2 · 9 Dec 2018

**Review for Hydrology and Earth System Sciences**

**Oxycline oscillations induced by internal waves in deep Lake Iseo**

**Manuscript Number: hess-2018-542**

**by Giulia Valerio et al.**

This is a paper on an important topic: a detailed analysis of the dynamics and potential impacts of oxycline oscillations in a deep meromictic lake. In addition, the lake in question, Lake Iseo, is one of the five major Italian pre-Alpine lakes all of which are more or less meromictic, and which are extremely important from an economic and recreational standpoint.

The author team is well-qualified to conduct this application as the group includes scholars who are among the best European physical modelers focusing on such systems (Pilotti & Valerio), as well as two of the top researchers on lake sediments (Lau & Hupfer).

The paper is technically sound and is a major contribution to characterizing the impact of internal waves on sediment-water exchange of solutes. Although, as referenced by this paper, previous work has been conducted on the influence of internal waves in holomictic lakes, much less attention has been paid to how such oscillations effect transport at the chemoclines of meromictic systems. Because the chemocline is persistent and marks the boundary between regions with very different chemistries, this study is a major step towards eventually predicting the transport of nutrients and contaminants from the sediments back into a meromictic lake's surface waters.

Although I recommend that this paper be published, I have two general suggestions that should be addressed prior to publication.

1. I think the paper goes into too much detail regarding the results. I would suggest that the authors tighten up the text (as well as the figures and tables) to make the paper easier to follow. For example, I think some of the tables could be placed into supplementary materials.

2. Although I had no problem understanding the content and organization of the text, the authors are not native English speakers as I found lots of awkward wordings as well as typos that were frustrating. Here are a sampling of some lines that illustrate my point:

**Line 67:**
**that under the internal wave motions of the deep oxycline, the contiguous sediments undego**

**Line 141:**
**measured internal oscillations. This required to identify the temporal evolution of the periodicity and**

In the following, aside from the repetition ("the one the one"), there should be a space between the units "m" and "s":
**Line 173:**
**of 5 ms$^{-1}$, whose spatial and temporal structure fit the one the one predicted by the eigenmodel for**

**Line 389:**
**there are large and periodic displacements of the oxycline. The oxycline typically oscillation in the**

**Line 406:**
**Accordingly, this works provide experimental and numerical evidence of a chemical gradient**

**Line 445:**
**basin. The analysis of its oscillations over a 3 days window provided the time series of the area**

In the following, note that the two panels of Fig. 10 are not labeled (a) and (b):
**Line 436:**
**conditions will be mainly located in the northern, southern and eastern sub-basins (see Fig. 10a),**

There are quite a few other small errors of this type. If the journal does not provide very strong copyediting, I would suggest that the authors do a spellcheck and ask an English-speaking colleague to copyedit the article to smooth and make corrections to the manuscript.

---

## Author Comment (AC1) · 16 Jan 2019

**Oxycline oscillations induced by internal waves in deep Lake Iseo**
*by Giulia Valerio, Marco Pilotti, Maximilian Peter Lau, Michael Hupfer*

**Author comment: as both reviewers advised, the manuscript will be sent out to professional language editing shortly. In order to meet review deadlines, these responses to the reviewer comments contain the recommended amendments to the manuscript content. Please note that all amendments may be subject to further improvement during the language editing process.**

**Response to the Referee's comments**

**Response to the comments of Anonymous Referee #1**

*R1-C1. This manuscript contains very interesting material from a well designed field programme and good displays of measurements that should warrant publication. However, the text at its current quality is not good enough for publication. I hope the authors consider my points and do a proper internal review before submitting an improved version. (Comments see below). The manuscript deals with observations of internal waves in Lake Iseo and the monitoring of the oxicline motion over sediment at depths around 95m. The varying redox conditions impact on the geochemical processes at the sediment. The impacted area was estimated to 3% of the total lake bed. At times, the oxicline moved synchronously with the thermocline V1H1 mode at one oscillation per day, but earlier in summer the oscillations of oxycline and thermocline were out of sync. The oxycline oscillated at 1 cycle in four days. Later in autumn, signals were not so clear but roughly the same fundamental oscillation period of close to 4 days can be seen at both depths. The results are fascinating; these waves out of sync at various depth; after seeing Figs. 4,5,6, the reader is interested in hearing an explanation but the results section requires persistence to work one's way though the data processing details. In the end, I did not find an explanation for the a-syncronous behaviour.*

Reply R1-C1. We thank the Reviewer for the positive feedback about our manuscript and the field programme. We agree that the text can be improved according to the Reviewer's suggestions, as will be detailed in the following responses. In particular, we reviewed the whole text, focusing our effort in the facilitation of the reading of the data processing session and in the clarification of the main results of the investigation. For this purpose, we shortened the Results section as detailed in the reply R1-C21. We also tried to explain more clearly the asynchronous oscillations through this paragraph:

"In the strongly stratified period (Aug-Sept), we occasionally observed a decoupled internal wave response at the different depths (see Fig. 5). By comparing the periodicity of the measured oscillations and that of the unforced modes (see Fig. 3), the thermocline appears to be dominated by a V1H1 motion (~ 1 day period), while the oxycline by a V2H1 motion (~ 2-3 day period). This suggests that both modes were excited by the wind, but at different energy levels along the water column. This decoupled response can be explained by the vertical structure of the modes detailed in Table S1. In correspondence of the summer stratification, the V1H1 amplitudes are almost vertically uniform (1.1 < $\xi_4/\xi_2$ < 2.3). Conversely, the V2H1 amplitude at the chemocline $\xi_4$ is up to 5.7 times larger than that of the thermocline $\xi_2$. Accordingly, when a second vertical mode is excited by wind, the larger vertical displacements occur in correspondence to the deeper interface. This vertical amplification may explain why V2H1 mode is dominant in the deeper waters and is instead weaker than the V1H1 daily signal around the thermocline".

Finally, the manuscript will be sent to a professional editing service to further improve the quality of the text.

**Oxycline oscillations induced by internal waves in deep Lake Iseo**
*by Giulia Valerio, Marco Pilotti, Maximilian Peter Lau, Michael Hupfer*

*R1-C2. Lines 271 and 272: should it be 1/4 * 1/day and 1/2* 1/day ?*

Reply R1-C2. We thank the Reviewer for noting this oversight: we modified the text accordingly.

*R1-C3. Table 2: how have the depths, which separate the layers, been chosen? This choice has a major impact on the results. No density profiles have been shown for the three selected periods and no comparison with the discrete layer structure has been shown.*

Reply R1-C3. We agree with the Reviewer that a clearer explanation of this point is necessary. In the revised version of section 2.3 of the manuscript we specified the way we defined the depths of the layer interfaces:

"Under this condition, the upper interface of the metalimnion ($Z_2$ in Table 2) was located at the depth of maximum gradient in temperature (thermocline), while the lower interface ($Z_3$ in Table 2) was set to a depth of 35 m, below which the vertical temperature gradient strongly weakens. […]we considered an additional deep layer separated from the hypolimnion by the chemocline at 95 m ($Z_4$ in Table 2), which is characterized by a 25 mg L$^{-1}$ step in density due to the higher concentration of dissolved salts."

Following the suggestion of the Reviewer, we also modified Figure 2 (see Figure 2-R below), to now show a comparison of the discrete layer structure with the density profiles for the four representative months. The density profiles were computed on the basis of monthly averaged temperature profiles and one conductivity profile by means of an empirical equation of state that can account for the vertical gradients in conductivity. In the text we made reference to this Figure after the description of the layered structure:

"The resulting discrete density structure well describes the density profiles computed from the profiles of conductivity and temperature (see Figure 2c).".

Finally, we modified Table 2 (see Table 2-R below) by reporting the density differences between the layers, which is the physical quantity that is used for the computation of the modal periods.

**Table 2-R**. Features of the first horizontal, first, second and third vertical modes in Lake Iseo during a one year period. The monthly-averaged layered structure used for the calculation includes the depth $Z_i$ of the upper interface of each ith layer, and its density difference with respect to the deepest layer, $\rho_i - \rho_4$.

| Time | Layered structure | | | | | | Periods of the H1 modes | | |
| | $Z_2$ | $Z_3$ | $Z_4$ | $\rho_1 - \rho_4$ | $\rho_2 - \rho_4$ | $\rho_3 - \rho_4$ | V1 | V2 | V3 |
| | | (m) | | | kgm$^{-3}$ | | | (hours) | |
| --- | --- | --- | --- | --- | --- | --- | --- | --- | --- |
| Jul-17 | 12.5 | 35.0 | 95.0 | 1.700 | 0.223 | 0.029 | 26.7 | 65.1 | 88.9 |
| Aug-17 | 15.0 | 35.0 | 95.0 | 1.741 | 0.204 | 0.03 | 24.1 | 60.3 | 90.3 |
| Sep-17 | 17.5 | 35.0 | 95.0 | 1.486 | 0.187 | 0.031 | 23.7 | 65.4 | 92.5 |
| Oct-17 | 20.0 | 35.0 | 95.0 | 1.049 | 0.192 | 0.032 | 25.9 | 69.2 | 94.1 |
| Nov-17 | 22.5 | 35.0 | 95.0 | 0.645 | 0.187 | 0.034 | 31.0 | 74.4 | 102.8 |
| Dec-17 | 35.0 | - | 95.0 | 0.214 | 0.032 | - | 43.8 | 82.3 | - |
| Jan-18 | 45.0 | - | 95.0 | 0.059 | 0.028 | - | 67.7 | 139.2 | - |
| Feb-18 | 55.5 | - | 95.0 | 0.046 | 0.027 | - | 71.2 | 177.3 | - |
| Mar-18 | - | - | 95.0 | 0.025 | - | - | 78.5 | - | - |
| Apr-18 | 7.5 | 35.0 | 95.0 | 0.200 | 0.071 | 0.027 | 69.3 | 112.3 | 153.5 |

| May-18 | 10.0 | 35.0 | 95.0 | 0.596 | 0.107 | 0.028 | 48.4 | 75.8 | 108.7 |
| June-18 | 12.5 | 35.0 | 95.0 | 1.072 | 0.131 | 0.028 | 33.6 | 69.1 | 101.3 |

**Figure 2-R.** (a) Vertical profile of temperature compensated conductivity (EC25) and dissolved oxygen (DO) measured on 10/04/2018 at the LS-N. (b) Vertical profile of temperature (T) and dissolved oxygen (DO) measured on 22/07/2017 at the LS-N. The circles and the crosses show the depth of the dissolved oxygen sensors at LS-S and LS-N, respectively. The dots and the squares show the depth of the temperature sensors at LS-N, with the squares indicating the high accuracy sensors. (c1-c4) Vertical profiles of monthly-averaged density (solid lines) during four months and corresponding discrete layered structure (dashed line) used in the modal model. The density profiles were computed on the basis of an empirical equation of state that relates density to conductivity and temperature, calibrated for Lake Iseo according to the algorithm by Moreira et al. (2016) (details in Scattolini, 2018).

[Figure]

*R1-C4. Figs. 4,5,6 would the projection of wind on the lake axis not be more instructive than speed and direction?*

Reply R1-C4. We only partly agree with the Reviewer. We agree on the usefulness of including the southerly component of the wind. In this way, Figures 4, 5, 6 report a quantity which clearly highlights the daily pattern of the wind and which is consistent with the wavelet analysis of the wind reported in Figure 3e. Though, we also believe that this quantity is as informative as wind speed and direction are. For example, when the wind blows northerly, the direction at LS-N is around 60°C due to the orientation of the northern valley, while when the wind blows southerly it is 180°C. Accordingly, the projection of these two winds at the same intensity in the S-N direction would falsely suggest a stronger southerly wind. To avoid this ambiguity, we modified Figures 4, 5, 6 by keeping wind speed and direction, but adding the projection of the wind along the lake axis as suggested by the Reviewer. In the following we report the modified figures (-R).

Oxycline oscillations induced by internal waves in deep Lake Iseo
*by Giulia Valerio, Marco Pilotti, Maximilian Peter Lau, Michael Hupfer*

**Figure 4-R.** Time series measured from 14 to 24 October 2017 of (a) wind speed and its southerly component, (b) wind direction at LS-N, followed by the spatial and temporal variation of (c) temperature between 5 and 25 m at LS-N and of (d) dissolved oxygen between 80 and 105 m at LS-S. Panel (e) compares the time series of DO measurements at LS-N and LS-S stations. In correspondence of each tick of the horizontal axis it is 00:00 o'clock.

[Figure]

**Figure 5-R.** Same as Fig. 4 but referred to the period 28 August – 07 September 2017.

[Figure]

**Figure 6-R.** Same as Fig. 4 but referred to the period 05 – 18 December 2017.

[Figure]

*R1-C5. Fig. 10 writing too small. add display of area vs depth for entire lake*

Reply R1-C5. In response to the reviewer we modified Figure 10 by increasing the font's dimension and adding the display of area vs depth for the entire lake (see Figure 10-R below).

Oxycline oscillations induced by internal waves in deep Lake Iseo
*by Giulia Valerio, Marco Pilotti, Maximilian Peter Lau, Michael Hupfer*

**Figure 10-R**. (b) Estimation of the area of the bottom sediments subjected to alternating redox conditions. The areas were computed by considering the oscillations of the oxycline over a 3-day long time window. The three colours make reference to the contribution of the southern (S, blue), northern (N, red) and eastern basin (E, yellow), as shown on the map. In the left panel (a), the area-depth curves indicate the cumulative area "a" of the bottom situated below a given water depth in the whole lake (W) and in each sub-basin (E, N, S). In the x-axis, the area "a" was normalized with the total area "A" of each basin. The grey shaded area marks the maximum and minimum vertical displacement of the 0.5 mgDO L$^{-1}$ recorded at LS-S, highlighting the area of the bottom where the oxycline fluctuates.

[Figure]

*R1-C6. Fig.2 caption Electrical conductivity is "temperature compensated" not "normalized"*

Reply R1-C6. We agree with the Reviewer and we modified the caption accordingly.

*R1-C7. Fig. 3 could indicate the periods covered in Figs 4,5,6*

Reply R1-C7. We agree with the Reviewer and we indicated these periods by means of 3 rectangles, as shown in the following revised version of Figure 3.

**Oxycline oscillations induced by internal waves in deep Lake Iseo**
*by Giulia Valerio, Marco Pilotti, Maximilian Peter Lau, Michael Hupfer*

**Figure 3.R**. (a) Time series of the vertical displacements of the 12°C isotherm (white line) measured at LS-N, superimposed by the interpolated temperature distribution between 0 and 40 m, and (b) the associated continuous wavelet transform for the period between 23 July and 21 November 2017. (c) Time series of the vertical displacements of the 0.5 mgDO L-1 isoline measured at LS-S, superimposed to the interpolated distribution of oxygen, and (d) the associated continuous wavelet transform. (e) Natural periods of V1H1, V2H1 and V3H1 mode superimposed by the continuous wavelet transform of the N-S component of the wind measured at LS-N. The grey shaded region on either ends indicate the cone of influence, where edge effects become important. The three rectangles with a black outline show the three periods analyzed in the following Figures 4-6.

[Figure]

*R1-C8. Table 3: line of V2H1 xsi indices are wrong*

Reply R1-C8. We thank the Reviewer for noting this misprint. The Table was corrected accordingly.

*R1-C9. line 505: J. Ilmberger is NOT Jorg Imberger*

Reply R1-C9. We thank the Reviewer for noting this misprint. The reference was corrected accordingly.

*R1-C10. For modes in lakes, the authors may also see these papers on Lake Constance: Boehrer 2000: modal response in a deep stratified lake: western Lake Constance Appt et al 2004: basin scale motion in upper Lake Constance*

Reply R1-C10. We thanks the Reviewer for suggesting these two papers which report experimental evidence of second vertical modes in a deep lake and are now included in the Discussion section.

*R1-C11. quite often the components of a sentence are not in the correct order: e.g. line 25 to just list one: "These basin-scale internal waves cause in the water layer between 85 and 105 m depth a fluctuation of the oxygen concentration between 0 and 3 mg L-1 that, due to the bathymetry of the lake, changes the redox condition at the sediment surface."*

Reply R1-C11. We agree that the Reviewer. The mentioned sentence was replaced with "These basin-scale internal waves cause a fluctuation of the oxygen concentration between 0 and 3 mg $L^{-1}$ in the water layer between 85 and 105 m depth, changing the redox condition at the sediment surface."

*R1-C12. there are unnecessary words: as example: the first sentence of the paper line 33 "The physical processes occurring at ... " in stead of "Physical processes at ..."*

Reply R1-C12. We agree with the Reviewer and we modified the expression accordingly. To improve the whole style of the paper and correct linguistic errors like this one, the manuscript will be sent to a professional editing service.

*R1-C13. inconsistent names in line 116: SS-1 and SS-2 refer to LS-N and LS-S on Fig.1 ?*

Reply R1-C13. We thank the Reviewer for noting this misprint. SS-1 was replaced with SS-N and SS-2 with SS-S, consistently with the name given to the northern and southern shore stations shown in Fig. 1.

*R1-C14. line 297 wrong reference Table 2 should be Table 3*

Reply R1-C14. We thank the Reviewer for noting this misprint. The reference was corrected accordingly.

*R1-C15. often narrative style: examples for the results section: examples for the results section: line 319 "For the discussion that will follow, it is worthy to underline that, ... line 334 "The analysis of the measured data previously shown suggests the presence of ..." line 360 "At this point, it is of interest to reflect upon, ..."*

Reply R1-C15. We agree with the Reviewer that the style of the paper can benefit from a linguistic revision. Therefore, we modified the text, limiting the use of a narrative style and we will send the manuscript to a professional editing service for a further improvement. Concerning the mentioned sentences:

**Oxycline oscillations induced by internal waves in deep Lake Iseo**
*by Giulia Valerio, Marco Pilotti, Maximilian Peter Lau, Michael Hupfer*
* * *
"For the discussion that will follow, it is worthy to underline that" was replaced with

"We emphasize that";

"The analysis of the measured data previously shown suggests the presence of both a V1H1 and a V2H1 response. The obtained numerical results allowed us to clarify the nature of these oscillations and extend the spatial information provided by local measurements." was replaced with

"The obtained numerical results allowed us to clarify the nature of the observed oscillations and extend the spatial information provided by local measurements.";

"At this point, it is of interest to reflect upon the reasons of the excitations of these motions." was deleted.

*R1-C16.difficult sentences: one example in lines 402-405: "In Lake Iseo, a depth variation of the mineralization process along the water column generates a gravity driven segregation with a density gradient between the oxic mixolimnion and the anoxic monimolimnion, which favours the occurrence of large baroclinic motions at the interface of these layers, even if the water column is thermally homogenous."*

Reply R1-C16. We agree with the Reviewer that the sentence is not clear enough and can be simplified. We modified the sentence as: "In Lake Iseo, the decomposition of organic matter and the dissolution of its end products have favoured the accumulation of solutes in the deeper waters since the 80ies. Accordingly, a stable density gradient is present between the oxic mixolimnion and the anoxic monimolimnion, contributing to keep the latter isolated from the water above. This condition allows the occurrence of large baroclinic motions at the interface of these layers, even if the water column is thermally homogenous". Similar difficult sentences were simplified throughout the manuscript.

*R1-C17.not optimal choice of words: line 167 " we thickened the grid ..." is there no better choice? line 235 "streching" ? is "widening" better? line 243 "strongly different"*

Reply R1-C17. We agree with the Reviewer on a different choice for these expressions. Accordingly:

"we thickened the grid" was replaced with "using a vertical grid";

"metalimnion stretching" was replaced with "metalimnion widening";

"the internal wave field is strongly different, as clearly highlighted by the much larger excursions of the oxycline (up to 20 m) and their longer periodicity" was replaced with "the internal wave field is instead dominated by the much larger excursions of the oxycline (up to 20 m) and by longer periodicities"

*R1-C18.line 468- 483 is this discussion material?*

Reply R1-C18. We agree with the Reviewer that this section does not primarily discuss our results. Instead, it should give the reader an idea of the implications that a new perspective on oxycline oscillations would entail. In response to the reviewer we markedly shortened the respective section without completely removing it, building a clearer connection to our results and finally melted them with the two last concluding sentences:

"Oxycline dynamics affect the lake sediments with implications for the redox-controlled biogeochemical processes therein. Redox-sensitive P release (Søndergaard, 2003) may be halted in

the sediment depths segment affected by oxycline oscillations and shift P towards redox-insensitive fractions (Parsons et al., 2017). However, the role of sediments as sink and source of P is known to be controlled by a suite diagenetic processes including P supply, microbial mineralization, as well as the interacting processes between iron and sulphur (Hupfer and Lewandowski, 2008), and the susceptibility of these processes to excursion in oxygen availability is less well understood. According to our calculations, an additional 3% of the sediment area is affected by such excursions. The monimolimnion of lake Iseo store the vast majority of the in-lake P (360t of 480t, April 2016, Lau et al., in preparation), indicating the relevance of P release from sediment below the oxycline. Therefore, it remains crucial to further explore the dynamics in redox forcing in sediments of perennially stratified lakes and the entailing implications for internal P cycling and biogeochemical turnover."

*R1-C19.line 330 " the amplitude's reduction" ... "the lake's bathymetry"*

Reply R1-C19. We agree with the Reviewer and we deleted the questionable use of the Saxon genitive

*R1-C20.line 144 "natural" modes? I know normal modes, or modes of the internal wave equation or Taylor - Goldstein eq.*

Reply R1-C20. The expression makes reference to the structure of the modes independently from the forcing. We agree with the Reviewer that "free modes" or "unforced modes" could be a better expression than "natural modes" and so we made use of these expressions throughout the revised manuscript.

*R1-C21.The presented material has the potential for a good paper, but the writing needs to be improved. My recommendation is to get the results and the discussion focussed on the message of the paper. Shorten the text of these sections considerably. Make a proper internal review and if necessary ask for language editing.*

Reply R1-C21. We agree with the Reviewer that the text can be improved. Therefore, we reviewed the whole text, shortening both the Results and Discussion section. In particular:

- we removed Table 3 and Figure 9 from the main paper and placing them in a supplementary material document;
- we considerably shortened the Results section by focusing the text on the more relevant information. For example, we removed L188-189, L209-211, L222-223, most of L227-232, L277-284, L308-310, L326-333, L334-341, L359-361, and we strongly synthetized the description of low-pass filtering analysis (L237-242, L246-251).
- we removed from the Discussion L458-465 regarding the contribution of the eastern basin, which is of minor interest.
- we shortened the discussion about the P fluxes L468-483 and the details on the computations of the northern and eastern area subjected to alternating redox conditions L446-452.

Finally, the manuscript will be sent to a professional editing service.

---

## Author Response (AR1)

Oxycline oscillations induced by internal waves in deep Lake Iseo
*by Giulia Valerio, Marco Pilotti, Maximilian Peter Lau, Michael Hupfer*

**Response to the Referee's comments**

**Response to the comments of Anonymous Referee #1**

*R1-C1. This manuscript contains very interesting material from a well designed field programme and good displays of measurements that should warrant publication. However, the text at its current quality is not good enough for publication. I hope the authors consider my points and do a proper internal review before submitting an improved version. (Comments see below). The manuscript deals with observations of internal waves in Lake Iseo and the monitoring of the oxycline motion over sediment at depths around 95m. The varying redox conditions impact on the geochemical processes at the sediment. The impacted area was estimated to 3% of the total lake bed. At times, the oxicline moved synchronously with the thermocline V1H1 mode at one oscillation per day, but earlier in summer the oscillations of oxycline and thermocline were out of sync. The oxycline oscillated at 1 cycle in four days. Later in autumn, signals were not so clear but roughly the same fundamental oscillation period of close to 4 days can be seen at both depths. The results are fascinating; these waves out of sync at various depth; after seeing Figs. 4,5,6, the reader is interested in hearing an explanation but the results section requires persistence to work one's way though the data processing details. In the end, I did not find an explanation for the a-syncronous behaviour.*

Reply R1-C1. We thank the Reviewer for the positive feedback about our manuscript and the field programme. We agree that the text can be improved according to the Reviewer's suggestions, as will be detailed in the following responses. In particular, we reviewed the whole text, focusing our effort in the facilitation of the reading of the data processing session and in the clarification of the main results of the investigation. For this purpose, we shortened the Results section as detailed in the reply R1-C21. We also tried to explain more clearly the asynchronous oscillations through this paragraph (L297-309 of the revised paper):

"During the strongly stratified period (Aug–Sept), we occasionally observed a decoupled internal wave response at the different depths (see Fig. 5). By comparing the periodicity of the measured oscillations and that of the unforced modes (see Fig. 3), the thermocline appears to be dominated by a V1H1 motion (~ 1–day period), while the oxycline is dominated by a V2H1 motion (~ 2–3–day period). This suggests that both modes were excited by the wind, but at different energy levels along the water column. This decoupled response can be explained by the vertical structure of the modes detailed in Table S1. During the summer stratification, the V1H1 amplitudes are nearly vertically uniform ($1.1 < \xi_4/\xi_2 < 2.3$). Conversely, the V2H1 amplitude at the chemocline $\xi_4$ is up to 5.7 times larger than that of the thermocline $\xi_2$. Accordingly, when a second vertical mode is excited by the wind, the larger vertical displacements occur at the deeper interface. This vertical amplification may explain why the V2H1 mode is dominant in the deeper waters and is in contrast weaker than the V1H1 daily signal around the thermocline."

Finally, the manuscript was revised by a professional editing service to further improve the quality of the text.

*R1-C2. Lines 271 and 272: should it be 1/4 * 1/day and 1/2* 1/day ?*

Reply R1-C2. We thank the Reviewer for noting this oversight: we modified the text accordingly.

**Oxycline oscillations induced by internal waves in deep Lake Iseo**
*by Giulia Valerio, Marco Pilotti, Maximilian Peter Lau, Michael Hupfer*
* * *
*R1-C3. Table 2: how have the depths, which separate the layers, been chosen? This choice has a major impact on the results. No density profiles have been shown for the three selected periods and no comparison with the discrete layer structure has been shown.*

Reply R1-C3. We agree with the Reviewer that a clearer explanation of this point is necessary. In the revised version of section 2.3 of the manuscript we specified the way we defined the depths of the layer interfaces (L146-156 of the revised paper):

"Under this condition, the upper interface of the metalimnion ($Z_2$ in Table 2) was set at the depth of the maximum temperature gradient (thermocline), while the lower interface ($Z_3$ in Table 2) was set at a depth of 35 m, below which the vertical temperature gradient strongly weakens. […] we considered an additional deep layer separated from the hypolimnion by the chemocline at 95 m depth ($Z_4$ in Table 2), which is characterized by a 25 mg $L^{-1}$ step in density because of the higher concentration of dissolved salts."

Following the suggestion of the Reviewer, we also modified Figure 2 (see Figure 2-R below), to now show a comparison of the discrete layer structure with the density profiles for the four representative months. The density profiles were computed on the basis of monthly averaged temperature profiles and one conductivity profile by means of an empirical equation of state that can account for the vertical gradients in conductivity. In the text we made reference to this Figure after the description of the layered structure (L158-159 of the revised paper):

"The resulting discrete density structure well describes the density profiles computed from the profiles of conductivity and temperature (see Figure 2c).".

Finally, we modified Table 2 (see Table 2-R below) by reporting the density differences between the layers, which is the physical quantity that is used for the computation of the modal periods.

**Table 2-R**. Features of the first horizontal, first, second and third vertical modes in Lake Iseo during a one year period. The monthly-averaged layered structure used for the calculation includes the depth $Z_i$ of the upper interface of each $i^{th}$ layer, and its density difference with respect to the deepest layer, $\rho_{i} - \rho_{4}$.

| Time | Layered structure | | | | | | Periods of the H1 modes | | |
|---|---|---|---|---|---|---|---|---|---|
| | $Z_2$ | $Z_3$ | $Z_4$ | $\rho_1 - \rho_4$ | $\rho_2 - \rho_4$ | $\rho_3 - \rho_4$ | V1 | V2 | V3 |
| | (m) | | | kgm$^{-3}$ | | | (hours) | | |
| Jul-17 | 12.5 | 35.0 | 95.0 | 1.700 | 0.223 | 0.029 | 26.7 | 65.1 | 88.9 |
| Aug-17 | 15.0 | 35.0 | 95.0 | 1.741 | 0.204 | 0.03 | 24.1 | 60.3 | 90.3 |
| Sep-17 | 17.5 | 35.0 | 95.0 | 1.486 | 0.187 | 0.031 | 23.7 | 65.4 | 92.5 |
| Oct-17 | 20.0 | 35.0 | 95.0 | 1.049 | 0.192 | 0.032 | 25.9 | 69.2 | 94.1 |
| Nov-17 | 22.5 | 35.0 | 95.0 | 0.645 | 0.187 | 0.034 | 31.0 | 74.4 | 102.8 |
| Dec-17 | 35.0 | - | 95.0 | 0.214 | 0.032 | - | 43.8 | 82.3 | - |
| Jan-18 | 45.0 | - | 95.0 | 0.059 | 0.028 | - | 67.7 | 139.2 | - |
| Feb-18 | 55.5 | - | 95.0 | 0.046 | 0.027 | - | 71.2 | 177.3 | - |
| Mar-18 | - | - | 95.0 | 0.025 | - | - | 78.5 | - | - |
| Apr-18 | 7.5 | 35.0 | 95.0 | 0.200 | 0.071 | 0.027 | 69.3 | 112.3 | 153.5 |
| May-18 | 10.0 | 35.0 | 95.0 | 0.596 | 0.107 | 0.028 | 48.4 | 75.8 | 108.7 |
| June-18 | 12.5 | 35.0 | 95.0 | 1.072 | 0.131 | 0.028 | 33.6 | 69.1 | 101.3 |

Oxycline oscillations induced by internal waves in deep Lake Iseo
*by Giulia Valerio, Marco Pilotti, Maximilian Peter Lau, Michael Hupfer*

**Figure 2-R.** (a) Vertical profile of temperature-compensated conductivity (EC25) and dissolved oxygen (DO) measured on 10/04/2018 at LS-N. (b) Vertical profile of temperature (T) and dissolved oxygen (DO) measured on 22/07/2017 at the LS-N. The circles and the crosses show the DO sensors at LS-S and LS-N, respectively. The dots and the squares show the depth of the temperature sensors at LS-N, with the squares indicating the high-accuracy sensors. (c1-c4) Vertical profiles of monthly averaged density (solid lines) during four months and the corresponding discrete layered structure (dashed line) used in the modal model. The density profiles were calculated on the basis of an empirical equation of state that relates density to conductivity and temperature, calibrated for lake Iseo according to the algorithm by Moreira et al. (2016) (details are in Scattolini, 2018).

[Figure]

*R1-C4. Figs. 4,5,6 would the projection of wind on the lake axis not be more instructive than speed and direction?*

Reply R1-C4. We only partly agree with the Reviewer. We agree on the usefulness of including the southerly component of the wind. In this way, Figures 4, 5, 6 report a quantity which clearly highlights the daily pattern of the wind and which is consistent with the wavelet analysis of the wind reported in Figure 3e. Though, we also believe that this quantity is as informative as wind speed and direction are. For example, when the wind blows northerly, the direction at LS-N is around 60°C due to the orientation of the northern valley, while when the wind blows southerly it is 180°C. Accordingly, the projection of these two winds at the same intensity in the S-N direction would falsely suggest a stronger southerly wind. To avoid this ambiguity, we modified Figures 4, 5, 6 by keeping wind speed and direction, but adding the projection of the wind along the lake axis as suggested by the Reviewer. In the following we report the modified figures (-R).

Oxycline oscillations induced by internal waves in deep Lake Iseo
*by Giulia Valerio, Marco Pilotti, Maximilian Peter Lau, Michael Hupfer*

**Figure 4-R.** Time series measured from the 14[th] to 24[th] of October 2017 of (a) wind speed and its southerly component and (b) wind direction at LS-N, followed by the spatial and temporal variation in (c) temperature between 5 and 25 m depth at LS-N and of (d) DO between 80 and 105 m at LS-S. Panel (e) compares the time series of DO measurements at the LS-N and LS-S stations. Corresponding to each tick of the horizontal axis it is the time 00:00.

[Figure]

**Figure 5-R.** Same as Fig. 4 but referring to the period of 28 August – 07 September 2017.

[Figure]

**Figure 6-R.** Same as Fig. 4 but referring to the period 05 – 18 December 2017.

[Figure]

*R1-C5.  Fig. 10 writing too small. add display of area vs depth for entire lake*

Reply R1-C5. In response to the reviewer we modified Figure 10 by increasing the font's dimension and adding the display of area vs depth for the entire lake (see Figure 10-R below).

Oxycline oscillations induced by internal waves in deep Lake Iseo
*by Giulia Valerio, Marco Pilotti, Maximilian Peter Lau, Michael Hupfer*

**Figure 10-R**. (b) Estimation of the area of the bottom sediments subjected to alternating redox conditions. The areas were calculated by considering the oscillations of the oxycline over a 3-day long time window. The three colours make reference to the contribution of the southern (S, blue), northern (N, red) and eastern basin (E, yellow), as shown on the map. In the left panel (a), the area-depth curves indicate the cumulative area "a" of the bottom situated below a given water depth in the whole lake (W) and in each sub-basin (E, N, S). On the x-axis, the area "a" was normalized with the total area "A" of each basin. The grey shaded area marks the maximum and minimum vertical displacement of the 0.5 mgDO L$^{-1}$ recorded at LS-S, highlighting the area of the bottom where the oxycline fluctuates.

[Figure]

*R1-C6. Fig.2 caption Electrical conductivity is "temperature compensated" not "normalized"*

Reply R1-C6. We agree with the Reviewer and we modified the caption accordingly.

*R1-C7. Fig. 3 could indicate the periods covered in Figs 4,5,6*

Reply R1-C7. We agree with the Reviewer and we indicated these periods by means of 3 rectangles, as shown in the following revised version of Figure 3.

**Oxycline oscillations induced by internal waves in deep Lake Iseo**
*by Giulia Valerio, Marco Pilotti, Maximilian Peter Lau, Michael Hupfer*

**Figure 3.R**. (a) Time series of the vertical displacements of the 12°C isotherm (white line) measured at LS-N, superimposed on the interpolated temperature distribution between 0 and 40 m, and (b) the associated continuous wavelet transform for the period between the 23rd July and 21st of November 2017. (c) Time series of the vertical displacements of the 0.5 mgDO L⁻¹ isoline measured at LS-S, superimposed on the interpolated distribution of oxygen, and (d) the associated continuous wavelet transform. (e) Natural periods of the V1H1, V2H1, and V3H1 modes superimposed on the continuous wavelet transform of the N-S component of the wind measured at LS-N. The grey shaded regions on either end indicate the cone of influence, where edge effects become important. The three rectangles with a black outline show the three periods analysed in the following Figures 4-6.

[Figure]

*R1-C8.  Table 3: line of V2H1 xsi indices are wrong*

Reply R1-C8. We thank the Reviewer for noting this misprint. The Table was corrected accordingly.

**Oxycline oscillations induced by internal waves in deep Lake Iseo**
*by Giulia Valerio, Marco Pilotti, Maximilian Peter Lau, Michael Hupfer*
* * *
*R1-C9.  line 505: J. Ilmberger is NOT Jorg Imberger*

Reply R1-C9. We thank the Reviewer for noting this misprint. The reference was corrected accordingly.

*R1-C10.For modes in lakes, the authors may also see these papers on Lake Constance: Boehrer 2000: modal response in a deep stratified lake: western Lake Constance Appt et al 2004: basin scale motion in upper Lake Constance*

Reply R1-C10. We thanks the Reviewer for suggesting these two papers which report experimental evidence of second vertical modes in a deep lake and are now included in the Discussion section.

*R1-C11.quite often the components of a sentence are not in the correct order: e.g. line 25 to just list one: "These basin-scale internal waves cause in the water layer between 85 and 105 m depth a fluctuation of the oxygen concentration between 0 and 3 mg L-1 that, due to the bathymetry of the lake, changes the redox condition at the sediment surface."*

Reply R1-C11. We agree that the Reviewer. The mentioned sentence was replaced with "These basin-scale internal waves cause a fluctuation in the oxygen concentration between 0 and 3 mg $L^{-1}$ in the water layer between 85 and 105 m depth, changing the redox condition at the sediment surface." (see L 25-17 of the revised paper)

*R1-C12.there are unnecessary words: as example: the first sentence of the paper line 33 "The physical processes occurring at ... " in stead of "Physical processes at ..."*

Reply R1-C12. We agree with the Reviewer and we modified the expression accordingly. To improve the whole style of the paper and correct linguistic errors like this one, the manuscript was revised by a professional editing service.

*R1-C13.inconsistent names in line 116: SS-1 and SS-2 refer to LS-N and LS-S on Fig.1 ?*

Reply R1-C13. We thank the Reviewer for noting this misprint. SS-1 was replaced with SS-N and SS-2 with SS-S, consistently with the name given to the northern and southern shore stations shown in Fig. 1.

*R1-C14.line 297 wrong reference Table 2 should be Table 3*

Reply R1-C14. We thank the Reviewer for noting this misprint. The reference was corrected accordingly.

*R1-C15.often narrative style: examples for the results section: examples for the results section: line 319 "For the discussion that will follow, it is worthy to underline that, ... line 334 "The analysis of the measured data previously shown suggests the presence of ..." line 360 "At this point, it is of interest to reflect upon, ..."*

**Oxycline oscillations induced by internal waves in deep Lake Iseo**
*by Giulia Valerio, Marco Pilotti, Maximilian Peter Lau, Michael Hupfer*

Reply R1-C15. We agree with the Reviewer that the style of the paper can benefit from a linguistic revision. Therefore, we modified the text, limiting the use of a narrative style and we sent the manuscript to a professional editing service for a further improvement. Concerning the mentioned sentences:

"For the discussion that will follow, it is worthy to underline that" was replaced with

"We emphasize that";

"The analysis of the measured data previously shown suggests the presence of both a V1H1 and a V2H1 response. The obtained numerical results allowed us to clarify the nature of these oscillations and extend the spatial information provided by local measurements." was replaced with

"The obtained numerical results clarify the nature of the observed oscillations and extend the spatial information provided by the local measurements.";

"At this point, it is of interest to reflect upon the reasons of the excitations of these motions." was deleted.

*R1-C16.difficult sentences: one example in lines 402-405: "In Lake Iseo, a depth variation of the mineralization process along the water column generates a gravity driven segregation with a density gradient between the oxic mixolimnion and the anoxic monimolimnion, which favours the occurrence of large baroclinic motions at the interface of these layers, even if the water column is thermally homogenous."*

Reply R1-C16. We agree with the Reviewer that the sentence is not clear enough and can be simplified. We modified the sentence as: "In Lake Iseo, the decomposition of organic matter and the dissolution of its end products have favoured the solutes accumulation in the deeper waters since the 1980s. Accordingly, a stable density gradient is present between the oxic mixolimnion and the anoxic monimolimnion, contributing to maintaining the latter isolated from the water above. This condition allows for the occurrence of large baroclinic motions at the interface of these layers, even if the water column is thermally homogenous." (see L352-357 of the revised paper) Similar difficult sentences were simplified throughout the manuscript.

*R1-C17.not optimal choice of words: line 167 " we thickened the grid ..." is there no better choice? line 235 "streching" ? is "widening" better? line 243 "strongly different"*

Reply R1-C17. We agree with the Reviewer on a different choice for these expressions. Accordingly:

"we thickened the grid" was replaced with "using a vertical grid"; (L166 of the revised paper)

"metalimnion stretching" was replaced with "metalimnion widening"; (L218; L227 of the revised paper)

"the internal wave field is strongly different, as clearly highlighted by the much larger excursions of the oxycline (up to 20 m) and their longer periodicity" was replaced with "the internal wave field is instead dominated by much larger excursions of the oxycline (up to 20 m) and longer periodicities." (L221-222 of the revised paper)

*R1-C18.line 468- 483 is this discussion material?*

Reply R1-C18. We agree with the Reviewer that this section does not primarily discuss our results. Instead, it should give the reader an idea of the implications that a new perspective on oxycline oscillations would entail. In response to the reviewer we markedly shortened the respective section without completely removing it, building a clearer connection to our results and finally melted them with the two last concluding sentences (L404-415 of the revised paper):

"Oxycline dynamics affect lake sediments with implications for the redox-controlled biogeochemical processes therein. Redox-sensitive P release (Søndergaard, 2003) may be halted in sediment depth segments affected by oxycline oscillations and shift P towards redox-insensitive fractions (Parsons et al., 2017). However, the role of sediments as sink and source of P is known to be controlled by a suite diagenetic processes including P supply, microbial mineralization, and interaction between iron and sulphur (Hupfer and Lewandowski, 2008). The susceptibility of these processes to excursion in oxygen availability is less well understood. According to our calculations, an additional 3% of the sediment area is affected by such excursions. The monimolimnion of Lake Iseo stores the vast majority of the in-lake P (360 t of 480 t, April 2016, Lau et al., in preparation), indicating the relevance of P release from sediment below the oxycline. Therefore, it remains crucial to further explore the dynamics in redox forcing in sediments of perennially stratified lakes and the entailing implications for internal P cycling and biogeochemical turnover."

*R1-C19. line 330 " the amplitude's reduction" ... "the lake's bathymetry"*

Reply R1-C19. We agree with the Reviewer and we deleted the questionable use of the Saxon genitive

*R1-C20. line 144 "natural" modes? I know normal modes, or modes of the internal wave equation or Taylor - Goldstein eq.*

Reply R1-C20. The expression makes reference to the structure of the modes independently from the forcing. We agree with the Reviewer that "free modes" or "unforced modes" could be a better expression than "natural modes" and so we made use of these expressions throughout the revised manuscript.

*R1-C21. The presented material has the potential for a good paper, but the writing needs to be improved. My recommendation is to get the results and the discussion focussed on the message of the paper. Shorten the text of these sections considerably. Make a proper internal review and if necessary ask for language editing.*

Reply R1-C21. We agree with the Reviewer that the text can be improved. Therefore, we reviewed the whole text, shortening both the Results and Discussion section. In particular:

- we removed Table 3 and Figure 9 from the main paper and placing them in a supplementary material document;
- we considerably shortened the Results section by focusing the text on the more relevant information. For example, we removed the following lined of the original paper: L188-189, L209-211, L222-223, most of L227-232, L277-284, L308-310, L326-333, L334-341, L359-361, and we strongly synthetized the description of low-pass filtering analysis (L237-242, L246-251 of the original paper).
- we removed from the Discussion at L458-465 of the original paper regarding the contribution of the eastern basin, which is of minor interest.
- we shortened the discussion about the P fluxes L468-483 of the original paper and the details on the computations of the northern and eastern area subjected to alternating redox conditions L446-452 of the original paper.

Oxycline oscillations induced by internal waves in deep Lake Iseo
*by Giulia Valerio, Marco Pilotti, Maximilian Peter Lau, Michael Hupfer*

Finally, the manuscript was revised by a professional editing service.

Oxycline oscillations induced by internal waves in deep Lake Iseo
*by Giulia Valerio, Marco Pilotti, Maximilian Peter Lau, Michael Hupfer*

**Response to the Referee's comments**

**Response to the comments of Anonymous Referee #2**

*R2-C1.This is a paper on an important topic: a detailed analysis of the dynamics and potential impacts of oxycline oscillations in a deep meromictic lake. In addition, the lake in question, Lake Iseo, is one of the five major Italian pre-Alpine lakes all of which are more or less meromictic, and which are extremely important from an economic and recreational standpoint. The author team is well-qualified to conduct this application as the group includes scholars who are among the best European physical modelers focusing on such systems (Pilotti & Valerio), as well as two of the top researchers on lake sediments (Lau & Hupfer). The paper is technically sound and is a major contribution to characterizing the impact of internal waves on sediment-water exchange of solutes. Although, as referenced by this paper, previous work has been conducted on the influence of internal waves in holomictic lakes, much less attention has been paid to how such oscillations effect transport at the chemoclines of meromictic systems. Because the chemocline is persistent and marks the boundary between regions with very different chemistries, this study is a major step towards eventually predicting the transport of nutrients and contaminants from the sediments back into a meromictic lake's surface waters. Although I recommend that this paper be published, I have two general suggestions that should be addressed prior to publication.*

Reply R2-C1. We sincerely thank the Reviewer for recognizing the importance and novelty of the topic.

*R2-C2.I think the paper goes into too much detail regarding the results. I would suggest that the authors tighten up the text (as well as the figures and tables) to make the paper easier to follow. For example, I think some of the tables could be placed into supplementary materials.*

Reply R2-C2. As also noted by the first reviewer, we agree with the Reviewer 2 that the paper can be made easier to follow. Therefore, we firstly followed the suggestion of the Reviewer, removing Table 3 and Figure 9 from the main paper and placing them in a supplementary material document. We also reviewed the whole text, shortening the Results section by focusing the text on the more relevant information. For example, we removed the following lines of the original paper: L188-189, L209-211, L222-223, most of L227-232, L277-284, L308-310, L326-333, L334-341, L359-361, and we strongly synthetized the description of low-pass filtering analysis (L237-242, L246-251 of the original paper)

*R2-C3.Although I had no problem understanding the content and organization of the text, the authors are not native English speakers as I found lots of awkward wordings as well as typos that were frustrating. Here are a sampling of some lines that illustrate my point. There are quite a few other small errors of this type. If the journal does not provide very strong copyediting, I would suggest that the authors do a spellcheck and ask an English-speaking colleague to copyedit the article to smooth and make corrections to the manuscript.*

**Oxycline oscillations induced by internal waves in deep Lake Iseo**
*by Giulia Valerio, Marco Pilotti, Maximilian Peter Lau, Michael Hupfer*

Reply R2-C3. We agree with the Reviewer that the manuscript can benefit from a linguistic revision. To improve the whole style of the paper and correct linguistic errors like the ones highlighted by the Reviewer, the manuscript was revised by a professional editing service.

*R2-C4.Line 67:that under the internal wave motions of the deep oxycline, the contiguous sediments undego*

Reply R2-C4. We thank the Reviewer for noting this misprint. "undego" was replaced with "undergo".

*R2-C5.Line 141:measured internal oscillations. This required to identify the temporal evolution of the periodicity and measured internal oscillations.*

Reply R2-C5. We agree with the Reviewer. The sentence was modified to "Therefore, we quantified the temporal evolution of the periodicity and the spatial structure of the free modes in Lake Iseo." (L136-137 of the revised paper)

*R2-C6.In the following, aside from the repetition ("the one the one"), there should be a space between the units "m" and "s": Line 173:of 5 ms-1, whose spatial and temporal structure fit the one the one predicted by the eigenmodel for*

Reply R2-C6. We thank the Reviewer for noting this misprint. The text was modified accordingly.

*R2-C7.Line 389:there are large and periodic displacements of the oxycline. The oxycline typically oscillation in the*

Reply R2-C7. We thank the Reviewer for noting this misprint. The sentence was modified as "The typical oxycline oscillation in the southern basin is in the range of 10 – 20 m, with periods ranging from 1 to 4 days." (l341-342 of the revised paper)

*R2-C8.Line 406:Accordingly, this works provide experimental and numerical evidence of a chemical gradient*

Reply R2-C8. We thank the Reviewer for noting this misprint. "this works provide" was replaced with "this work provides". (L357 of the revised paper)

*R2-C9.Line 445:basin. The analysis of its oscillations over a 3 days window provided the time series of the area*

Reply R2-C9. We agree with the Reviewer. The sentence is not present anymore in the main text, but in the caption of Figure 10. Here the sentence was modified as "The areas were calculated by considering the oscillations of the oxycline over a 3-day long time window." (L597-598 of the revised paper)

Oxycline oscillations induced by internal waves in deep Lake Iseo
*by Giulia Valerio, Marco Pilotti, Maximilian Peter Lau, Michael Hupfer*

*R2-C10.In the following, note that the two panels of Fig. 10 are not labeled (a) and (b): 436:conditions will be mainly located in the northern, southern and eastern sub-basins (see Fig. 10a).*

Reply R2-C10. The panels were labelled but with a small font size. Accordingly, the figure was modified and a) and b) were written in a larger font (see Figure 10-R below).

Oxycline oscillations induced by internal waves in deep Lake Iseo
*by Giulia Valerio, Marco Pilotti, Maximilian Peter Lau, Michael Hupfer*

**Figure 10-R**. (b) Estimation of the area of the bottom sediments subjected to alternating redox conditions. The areas were calculated by considering the oscillations of the oxycline over a 3-day long time window. The three colours make reference to the contribution of the southern (S, blue), northern (N, red) and eastern basin (E, yellow), as shown on the map. In the left panel (a), the area-depth curves indicate the cumulative area "a" of the bottom situated below a given water depth in the whole lake (W) and in each sub-basin (E, N, S). On the x-axis, the area "a" was normalized with the total area "A" of each basin. The grey shaded area marks the maximum and minimum vertical displacement of the 0.5 mgDO L$^{-1}$ recorded at LS-S, highlighting the area of the bottom where the oxycline fluctuates.

[Figure]

.

Oxycline oscillations induced by internal waves in deep Lake Iseo
*by Giulia Valerio, Marco Pilotti, Maximilian Peter Lau, Michael Hupfer*

**List of all relevant changes made in the mauscript**

- Whole revision of text, focusing our effort in the facilitation of the reading of the data processing session and in the clarification of the main results of the investigation.
- Shortening of both the Results and Discussion section.
- Removal of Table 3 and Figure 9 from the main paper, placing them in a supplementary material document.
- Linguistic revision of the text by a professional editing service
- More detailed analysis of the layered structure used in the model, including a graphical comparison of the discrete layer structure with the density profiles for the four representative months (see changes in Table 2 and Figure 2)
- Modification of Figure 10 of the original manuscript (now Figure 9) with a more effective description of the area-depth curves

[revised manuscript text omitted]